# Human T cell receptor occurrence patterns encode immune history, genetic background, and receptor specificity

William S DeWitt III[1,2], Anajane Smith[3], Gary Schoch[3], John A Hansen[3,4], Frederick A Matsen IV[1,2], Philip Bradley[1,5]*

[1]Public Health Sciences Division, Fred Hutchinson Cancer Research Center, Seattle, United States; [2]Department of Genome Sciences, University of Washington, Seattle, United States; [3]Clinical Division, Fred Hutchinson Cancer Research Center, Seattle, United States; [4]Department of Medicine, University of Washington, Seattle, United States; [5]Institute for Protein Design, University of Washington, Seattle, United States

**Abstract** The T cell receptor (TCR) repertoire encodes immune exposure history through the dynamic formation of immunological memory. Statistical analysis of repertoire sequencing data has the potential to decode disease associations from large cohorts with measured phenotypes. However, the repertoire perturbation induced by a given immunological challenge is conditioned on genetic background via major histocompatibility complex (MHC) polymorphism. We explore associations between MHC alleles, immune exposures, and shared TCRs in a large human cohort. Using a previously published repertoire sequencing dataset augmented with high-resolution MHC genotyping, our analysis reveals rich structure: striking imprints of common pathogens, clusters of co-occurring TCRs that may represent markers of shared immune exposures, and substantial variations in TCR-MHC association strength across MHC loci. Guided by atomic contacts in solved TCR:peptide-MHC structures, we identify sequence covariation between TCR and MHC. These insights and our analysis framework lay the groundwork for further explorations into TCR diversity.
DOI: https://doi.org/10.7554/eLife.38358.001

*For correspondence:
pbradley@fredhutch.org

Competing interests: The authors declare that no competing interests exist.

## Introduction

T cells are the effectors of cell-mediated adaptive immunity in jawed vertebrates. To control a broad array of pathogens, massive genetic diversity in loci encoding the T cell receptor (TCR) is generated somatically throughout an individual's life via a process called V(D)J recombination. All nucleated cells regularly process and present internal peptide antigens on cell surface molecules called major histocompatibility complex (MHC). Through the interface of TCR and MHC, a T cell with a TCR having affinity for a peptide antigen complexed with MHC (pMHC) is stimulated to initiate an immune response to an infected (or cancerous) cell. The responding T cell proliferates clonally, and its progeny inherit the same antigen-specific TCR, constituting long-term immunological memory of the antigen. The diverse population of TCR clones in an individual (the TCR repertoire) thus dynamically encodes a history of immunological challenges.

Advances in high-throughput TCR sequencing have shown the potential of the TCR repertoire as a personalized diagnostic of pathogen exposure history, cancer, and autoimmunity (*Thomas et al., 2014*; *Kirsch et al., 2015*; *Friedensohn et al., 2017*; *Ostmeyer et al., 2017*). Public TCRs—defined as TCR sequences seen in multiple individuals and perhaps associated with a shared disease phenotype—have been found in a range of infectious and autoimmune diseases and cancers including influenza, Epstein-Barr virus, and cytomegalovirus infections, type I diabetes, rheumatoid arthritis,

**eLife digest** The immune system has two major ways of clearing up an infection. A rapid, first line of defense buys time while the second 'adaptive' response disposes of the threat with precision. The adaptive response takes longer to develop but once it has dealt with a disease, it remembers: the next time the body encounters the same threat, the immune system can respond much faster.

When cells are infected by a disease-causing microbe, like a bacterium or a virus, they start carrying fragments of that microbe on their surface. Immune cells known as T cells then recognize these fragments using proteins called T cell receptors. Each T cell has a different receptor, which is specific to a precise fragment of a particular microbe. After successfully clearing an infection, some of the T cells that were mobilized remain in the blood. These memory T cells, and their specific receptors, are a lasting trace of the infections a person has encountered in the past.

The exact portion of the microbial fragments that the T cells receptors can 'see' depends on another set of proteins, called MHC. These hold the fragments at the surface of the infected cells. The genes that code for MHCs are incredibly diverse, to the point that the exact combination of MHCs carried by a cell can be specific to an individual. However, different MHCs present different microbial fragments, and this changes which receptor can recognize the infection. At the level of a population, this mechanism makes it difficult to use T cell receptors to know exactly which diseases people had to face.

Here, DeWitt et al. look at the T cell receptor sequences of 666 healthy participants, as well as their MHC variants, to try to reconstruct their disease history. This revealed that many people have clusters of similar T cells receptors sequences that occur together; these could be linked to exposure to common viruses such as parvovirus, influenza, cytomegalovirus and Epstein-Barr virus. Furthermore, examining 3D structures of T cell receptors binding to fragments carried by MHCs helps to identify how changes in the sequence of the MHC can influence which receptor will be able to attach to the complex.

These results show that, despite the diversity and complexity of T cell receptors and MHCs, it is possible to spot patterns across people, and to start understanding how those patterns emerge. In addition to fighting body invaders, T cells can also use their receptors to recognize certain protein fragments carried by tumor cells. Improving our knowledge of T cell receptors and MHCs could give new insights to fight cancer.

DOI: https://doi.org/10.7554/eLife.38358.002

and melanoma (*Venturi et al., 2008*; *Li et al., 2012*; *Madi et al., 2017*; *Pogorelyy et al., 2017*; *Dash et al., 2017*; *Glanville et al., 2017*; *Chu et al., 2018*; *Pogorelyy et al., 2018*). By correlating occurrence patterns of public TCR$\beta$ chains with cytomegalovirus (CMV) serostatus across a large cohort of healthy individuals, Emerson et al. identified a set of CMV-associated TCR chains whose aggregate occurrence was highly predictive of CMV seropositivity (*Emerson et al., 2017*). Staining with multimerized pMHC followed by flow cytometry has been used to isolate and characterize large populations of T cells that bind to defined pMHC epitopes (*Dash et al., 2017*; *Glanville et al., 2017*), providing valuable data on the mapping between TCR sequence and epitope specificity. We and others have leveraged these data to develop learning-based models of TCR:pMHC interactions, using TCR distance measures (*Dash et al., 2017*), CDR3 sequence motifs (*Glanville et al., 2017*) and k-mer frequencies (*Cinelli et al., 2017*), and other techniques.

MHC proteins in humans are encoded by the human leukocyte antigen (HLA) loci and are among the most polymorphic in the human genome (*Robinson et al., 2015*). Within an individual, six major antigen-presenting proteins are each encoded by polymorphic alleles. The set of these alleles comprise the individual's HLA type, which is unlikely to be shared with an unrelated individual and which determines the subset of peptide epitopes presented to T cells for immune surveillance. Specificity of a given TCR for a given antigen is biophysically modulated by MHC structure: MHC binding specificity determines the specific antigenic peptide that is presented, and the TCR binds to a hybrid molecular surface composed of peptide- and MHC-derived residues. Thus, population-level studies of TCR-disease association are severely complicated by a dependence on individual HLA type.

Here we report an analysis of the occurrence patterns of public TCRs in a cohort of 666 healthy volunteer donors, in which information on only TCR sequence and HLA association guide us to inferences concerning disease history. To complement deep TCR$\beta$ repertoire sequencing available from a previous study (*Emerson et al., 2017*), we have assembled high-resolution HLA typing data at the major class I and class II HLA loci on the same cohort, as well as information on age, sex, ethnicity, and CMV serostatus. We focus on statistical association of TCR occurrence with HLA type, and show that many of the most highly HLA-associated TCRs are likely responsive to common pathogens: for example, eight of the ten TCR$\beta$ chains most highly associated with the HLA-A*02:01 allele are likely responsive to one of two viral epitopes (influenza M1$_{58}$ and Epstein-Barr virus BMLF1$_{280}$). We introduce new approaches to cluster TCRs by primary sequence and by the pattern of occurrences among individuals in the cohort, and we identify highly significant TCR clusters that may indicate markers of immunological memory. Four of the top five most significant clusters appear linked with common pathogens (parvovirus B19, influenza virus, CMV, and Epstein-Barr virus), again highlighting the impact of viral pathogens on the public repertoire. We also find HLA-unrestricted TCR clusters, some likely to be mucosal-associated invariant T (MAIT) cells, which recognize bacterial metabolites presented by non-polymorphic MR1 proteins, rather than pMHC (*Kjer-Nielsen et al., 2012*). Our global analysis of TCR-HLA association identifies striking variation in association strength across HLA loci and highlights trends in V(D)J generation probability and degree of clonal expansion that illuminate selection processes in cellular immunity. Guided by structural analysis, we used our large dataset of HLA-associated TCR$\beta$ chains to identify statistically significant sequence covariation between the TCR CDR3 loop and the DRB1 allele sequence that preserves charge complementarity at the TCR:pMHC interface. These analyses help elucidate the complex dependence of TCR sharing on HLA type and immune exposure, and will inform the growing number of studies seeking to identify TCR-based disease diagnostics.

## Results

### The matrix of public TCRs

Of the 80 million unique TCR$\beta$ chains (defined by V-gene family and CDR3 sequence) in the 666 cohort repertoires, about 11 million chains are found in at least two individuals and referred to here as *public* chains (for a more nuanced examination of TCR chain sharing see [*Elhanati et al., 2018*]). The occurrence patterns of these public TCR$\beta$s—the subset of subjects in which each distinct chain occurs—can be thought of as forming a very large binary matrix $M$ with about 11 million rows and 666 columns. Entry $M_{i,j}$ contains a one or a zero indicating presence or absence, respectively, of TCR $i$ in the repertoire of subject $j$ (ignoring for the moment the abundance of TCR $i$ in repertoire $j$; *Figure 1* depicts two illustrative sub-matrices of $M$). (*Emerson et al., 2017*) demonstrated that this binary occurrence matrix $M$ encodes information on subject genotype and immune history: they were able to successfully predict HLA-A and HLA-B allele type and CMV serostatus by learning sets of public TCR$\beta$ chains with occurrence patterns that were predictive of these features. Specifically, each feature—such as the presence of a given HLA allele (e.g. HLA-A*02:01) or CMV seropositivity—defines a subset of the cohort members positive for that feature, and can be encoded as a vector of 666 binary digits. This phenotype occurrence pattern of zeros and ones can be compared to the occurrence patterns of all the public TCR$\beta$ chains to identify similar patterns, as quantified by a *p*-value for significance of co-occurrence across the 666 subjects; thresholding on this *p*-value produces a subset of significantly associated TCR$\beta$ chains whose collective occurrence in a repertoire was found by Emerson et al. to be predictive of the feature of interest (in cross-validation and, for CMV, on an independent cohort). Generalizing from these results, it is reasonable to expect that other common immune exposures may be encoded in the occurrence matrix $M$, and that these encodings could be discovered if we had additional phenotypic data to correlate with TCR occurrence patterns. In this study, we set out to discover these encoded exposures de novo, without additional phenotypic correlates, by learning directly from the structure of the occurrence matrix $M$ and using as well the sequences of the TCR$\beta$ chains (both their similarities to one another and to TCR sequences characterized in the literature). We hypothesized that patterns of TCR co-occurrence (correlations between rows in the matrix $M$) might indicate shared responses to unknown immune exposures, that co-occurrence between TCR chains and HLA alleles (correlations between rows in $M$ and rows in the

HLA allele occurrence matrix) could be used to help identify functional TCR chains, and that clustering TCR$\beta$ chains by co-occurrence and sequence could highlight functional associations (*Figure 1*). To support this effort we assembled additional HLA typing data for the subjects, now at 4-digit resolution (e.g., A*02:01 rather than A*02) and including MHC class II alleles, and we compiled a dataset of annotated TCR$\beta$ chains by combining online TCR sequence databases, structurally characterized TCRs, and published studies (see Materials and methods; [*Shugay et al., 2018*; *Tickotsky et al., 2017*; *Berman et al., 2000*; *Dash et al., 2017*; *Glanville et al., 2017*; *Song et al., 2017*; *Kasprowicz et al., 2006*]). Here we describe the outcome of this discovery process, and we report a number of intriguing general observations about the role of HLA in shaping the T cell repertoire.

The results of our analysis are organized in the remaining five sections as follows. We begin with an examination of TCR co-occurrence patterns across the full cohort (first section, Figures 2–3). In the next section we examine patterns of TCR-HLA association (Table 1 and Figures 4–5). In the third section we analyze TCR co-occurrence within subsets of the cohort positive for specific HLA alleles, and we identify TCR clusters that may be reflective of shared immune exposures (Figures 6–8). In the fourth section we use our dataset of HLA-associated TCR$\beta$ chains to identify covariation between HLA and the TCR$\beta$ CDR3 sequence (Table 2 and Figure 9). In the final section we focus on CMV-responsive TCR$\beta$ chains, examining their degree of HLA-restriction and the extent to which they may be responding to other antigens (Figure 10). *Figure 1* provides a graphical overview of the co-occurrence analysis.

## Globally co-occurring TCR pairs form clusters defined by shared associations

We hypothesized that we could identify unknown immune exposures encoded in the public repertoire by comparing the occurrence patterns of individual TCR$\beta$ chains to one another. A subset of TCR$\beta$ chains that strongly co-occur across the 666 cohort subjects might correspond to an unmeasured immune exposure that is common to a subset of subjects. Since shared HLA restriction could represent an alternative explanation for significant TCR co-occurrence, we also compared the TCR occurrence patterns to the occurrence patterns for class I and class II HLA alleles. We began by analyzing TCR occurrence patterns over the full set of cohort members. For each pair of public TCR$\beta$ chains $t_1$ and $t_2$ we computed a co-occurrence $p$-value $P_{\mathrm{CO}}(t_1, t_2)$ that reflects the probability of seeing an equal or greater overlap of shared subjects (i.e., subjects in whose repertoires both $t_1$ and $t_2$ are found) if the occurrence patterns of the two TCRs had been chosen randomly (for details, see Materials and methods and Figure 12). In a similar manner we computed, for each HLA allele $a$ and TCR $t$, an association $p$-value $P_{\mathrm{HLA}}(a, t)$ that measures the degree to which TCR $t$ tends to occur in subjects positive for allele $a$. Finally, for each pair of strongly co-occurring ($P_{\mathrm{CO}} < 1 \times 10^{-8}$) TCR$\beta$ chains $t_1$ and $t_2$, we looked for a mutual HLA association that might explain their co-occurrence, by finding the allele having the strongest association with both $t_1$ and $t_2$, and noting its association $p$-value:

$$P_{\mathrm{HLA}}(t_1, t_2) = \min_{a \in A} \max_{t \in \{t_1, t_2\}} P_{\mathrm{HLA}}(a, t),$$

where $A$ denotes the set of all HLA alleles. In words, we take the $p$-value of the strongest HLA allele association with the TCR pair, where the association of an HLA allele with a TCR pair is defined by the weakest association of the allele among the individual TCRs.

Based on this analysis, we identified two broad classes of strongly co-occurring TCR pairs (*Figure 2*): those with a highly significant shared HLA association, where the co-occurrence of the two TCRs can be explained by a shared HLA allele association (i.e. a common HLA restriction), and those with only modest shared HLA-association $p$-value, for which another explanation of co-occurrence must be sought. Points above the dashed $y = x$ line correspond to pairs of TCRs for which there exists an HLA allele whose co-occurrence with each of the TCRs is stronger than their mutual co-occurrence, while for points below the line no such HLA allele was present in the dataset.

We used a neighbor-based clustering algorithm, DBSCAN (*Ester et al., 1996*), to link strongly co-occurring TCR pairs together to form larger correlated clusters (see Materials and methods), and then investigated phenotype associations with these clusters. At an approximate family-wise error rate of $0.05$ (see Materials and methods), we identified 28 clusters of co-occurring TCRs, with sizes ranging from 7 to 386 TCRs (*Figure 3*). Given one of these clusters of co-occurring TCRs, we can

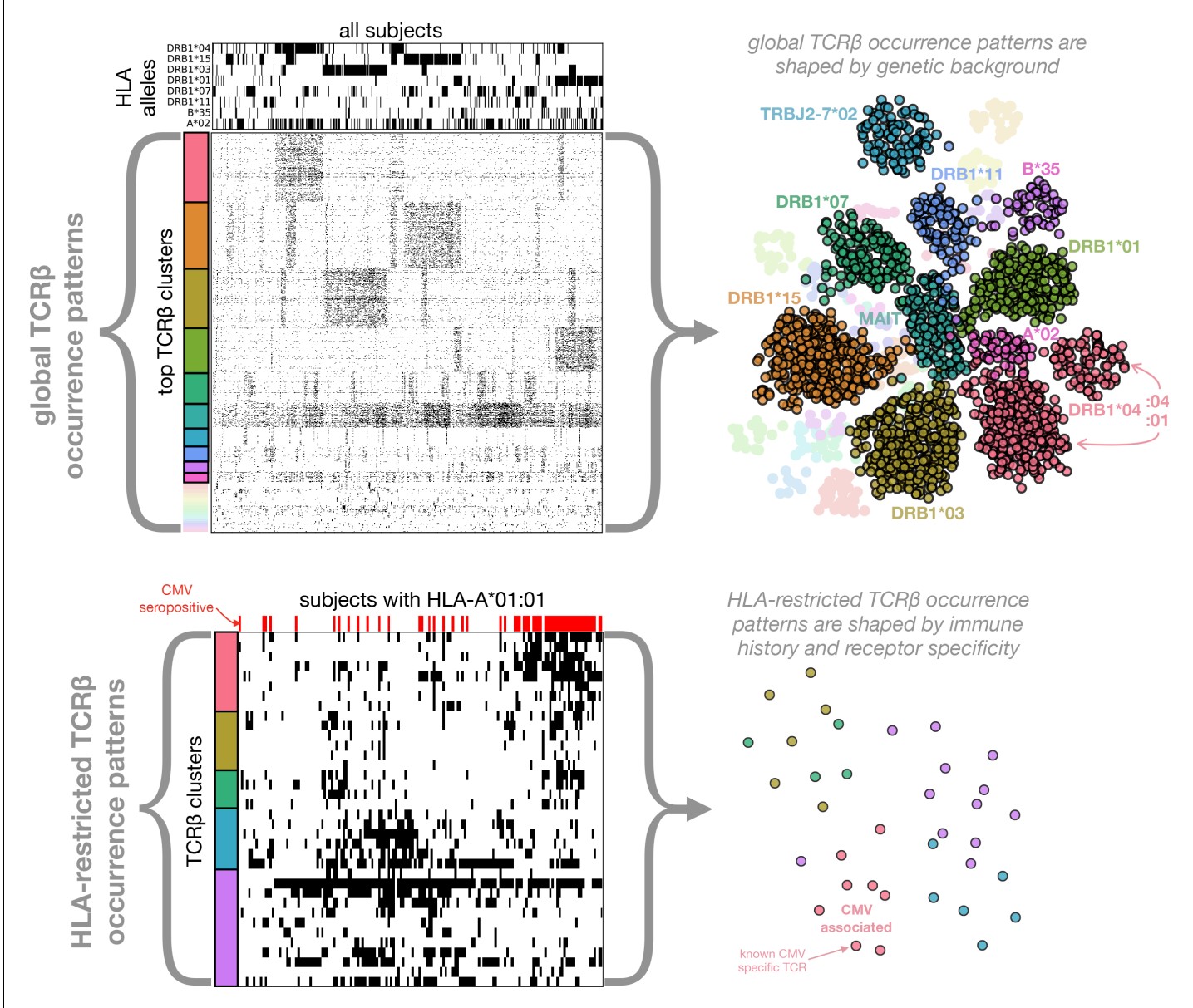

**Figure 1.** Clustering of TCR occurrence patterns across the full cohort (top) and within a cohort subset defined by a shared HLA allele (bottom). As described in detail in the following sections, we used covariation analysis to identify clusters of co-occurring TCRβ chains. Here we provide a graphical introduction to these results by depicting occurrence patterns of clustered TCRs over the full cohort and over a cohort subset defined by a single HLA allele (HLA-A*01:01). TCR clusters over the full cohort are largely driven by the occurrence patterns of specific HLA alleles (compare the occurrence patterns of the top five global clusters to those of the top 5 HLA alleles, respectively), whereas HLA-restricted clusters may reflect shared immune exposures, as illustrated here by a CMV-associated TCR cluster (the pink cluster in the bottom panels). In the top left panels, occurrence patterns of HLA alleles and TCRβ chains (rows) are indicated for each of the cohort subjects (columns) by filled (black) matrix elements. The TCRβ chains chosen for depiction in the occurrence matrix are the members of the 28 global co-occurrence clusters identified in section 'Globally co-occurring TCR pairs form clusters defined by shared associations'. The TCRs (rows) are ordered by cluster membership as indicated by colored bands to the left of the matrix. The selected HLA alleles correspond to the strongest associations for the top 10 clusters (two of which are not HLA-associated). The cohort subjects are ordered by column similarity so as to emphasize block structure present in the matrix. The bottom left panels similarly show occurrence patterns for HLA-A*01:01-associated TCRβ chain clusters over the subset of subjects carrying this allele, alongside an indicator of cytomegalovirus seropositivity for each subject (red). In-depth analysis of these (and other) HLA-associated TCRβ clusters is presented in section 'HLA-restricted TCR clusters'. For visualization purposes, two-dimensional embeddings of the TCRβ chains based on their occurrence patterns (binary strings representing presence/absence in the subjects) are depicted in the right panels, with the TCR chains colored by cluster assignment and annotated by known associations.

DOI: https://doi.org/10.7554/eLife.38358.003

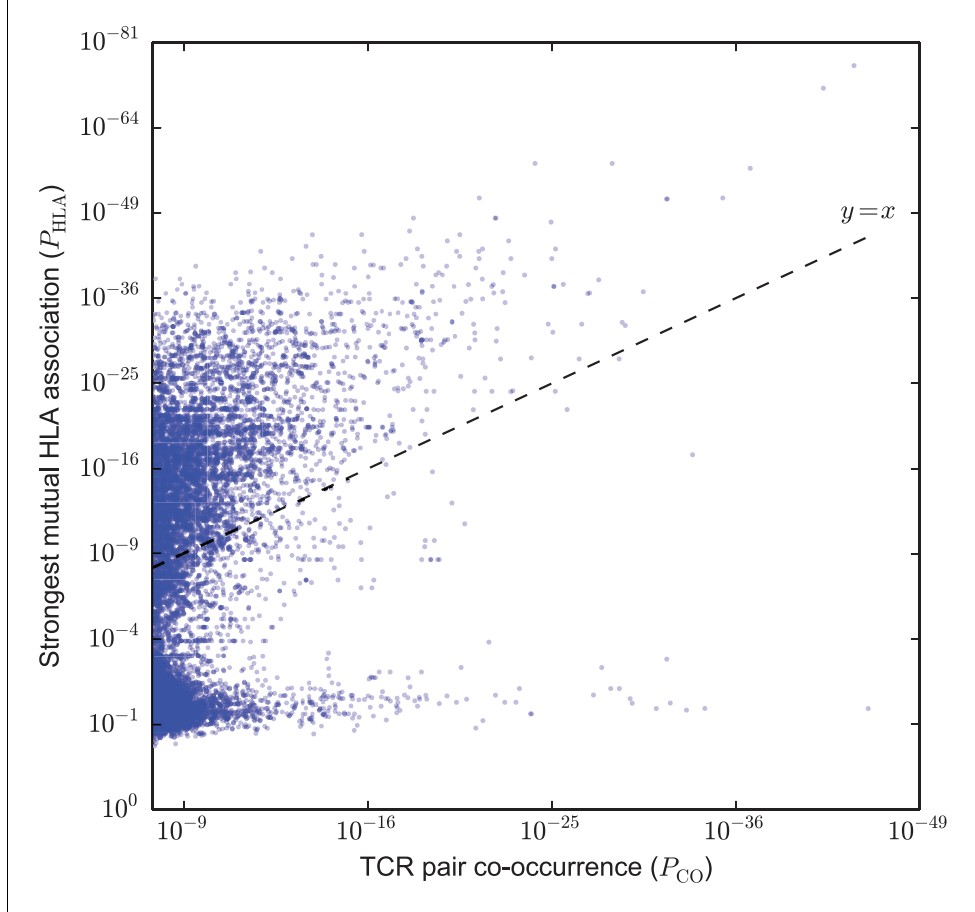

**Figure 2.** Strongly co-occurring TCR pairs form two broad classes distinguished by HLA-association strength. The co-occurrence $p$-value $P_{CO}$ for each pair of public TCRs is plotted ($x$-axis) against the HLA-association $p$-value $P_{HLA}$ for the HLA allele with the strongest mutual association with that TCR pair ($y$-axis). There are 6092 TCR-pairs above the diagonal ($y = x$) and 4713 pairs below the diagonal.

DOI: https://doi.org/10.7554/eLife.38358.004

The following source data is available for figure 2:

**Source data 1.** TCR pairs and corresponding $P_{CO}$ and $P_{HLA}$ values.

DOI: https://doi.org/10.7554/eLife.38358.005

count the number of cluster member TCRs found in each subject's repertoire. The aggregate occurrence pattern of the cluster can be visualized as a rank plot of this cluster TCR count over the subjects (the black curves in *Figure 3C–D*). This ranking can also be compared with other phenotypic or genotypic features of the same subjects. In particular, by comparing this aggregate occurrence pattern to a control pattern generated by repeatedly choosing equal numbers of subjects independently at random (dotted green lines in *Figure 3C–D*), we can identify a subset of the cohort with an apparent enrichment of cluster member TCRs and look for overlap between this subset and other defined cohort features. Performing this comparison against the occurrence patterns of class I and class II HLA alleles revealed that the majority of the TCR clusters were strongly associated with at least one HLA allele (as depicted for a DRB1*15:01-associated cluster in *Figure 3C* and summarized in *Figure 3B*).

In addition, there were two large clusters of TCRs which were not strongly associated with any of the typed HLA alleles (clusters 6 and 7 in *Figure 3*). Visual inspection of the CDR3 regions of TCRs in one of these clusters revealed a distinctive 'YV' C-terminal motif that is characteristic of the TRBJ2-7*02 allele (*Figure 3—figure supplement 1*), and indeed the 41 subjects whose repertoires indicated the presence of this genetic variant were exactly the 41 subjects enriched for members of this TCR cluster (*Figure 3D*). This demonstrated that population diversity in germline allele sets

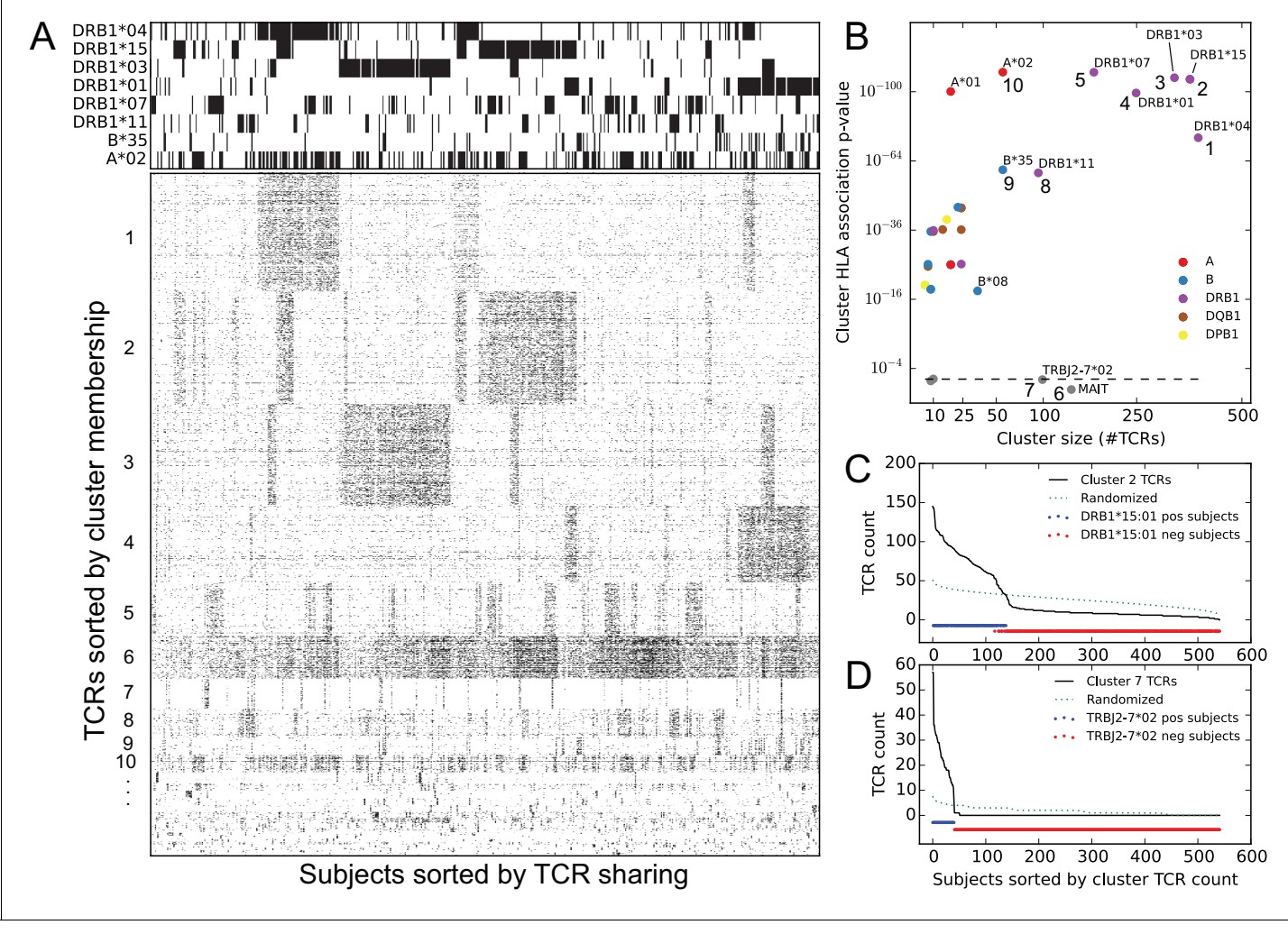

**Figure 3.** Clustering public TCRβ chains by co-occurrence over the full cohort identifies associations with HLA and TRBJ alleles as well as an invariant T cell subset. (**A**) Graphical representations of the TCRβ chain occurrence matrix (lower left) and the HLA-allele occurrence matrix (upper left), restricted to members of the 28 global co-occurrence TCR clusters and the associated HLA alleles for the top 10 clusters, respectively. TCRβ chains (rows) are ordered by cluster membership and subjects (columns) are ordered by column similarity (Jaccard distance of TCR sets) to emphasize block structure present in the matrix. (**B**) Cluster size (*x*-axis) versus the *p*-value of the most significant HLA allele association (*y*-axis), with markers colored according to the locus of the associated allele. Dashed line indicates random expectation based on the total number of alleles, assuming independence. (**C**) Count of cluster member TCRs found in each subject for the cluster labeled '2' in panel (**B**) (top right). The dotted line represents an averaged curve based on randomly and independently selecting subject sets for each member TCR. Red and blue dots indicate the occurrence of the DRB1*15:01 allele in the cohort. (**D**) Count of cluster member TCRs found in each subject for the cluster labeled '7' in panel (**B**) (center bottom). The dotted line again represents a control pattern, and the red and blue dots indicate the occurrence of the TRBJ2-7*02 allele.

DOI: https://doi.org/10.7554/eLife.38358.006

The following source data and figure supplements are available for figure 3:

**Source data 1.** Cluster sizes and HLA-allele association *p*-values.
DOI: https://doi.org/10.7554/eLife.38358.010
**Figure supplement 1.** TCRdist tree of the members of the TRBJ2-7*02-associated cluster.
DOI: https://doi.org/10.7554/eLife.38358.007
**Figure supplement 2.** TCRdist tree of the members of the putative MAIT cell cluster.
DOI: https://doi.org/10.7554/eLife.38358.008
**Figure supplement 3.** More details on the MAIT cell cluster: subject age and N-nucleotide insertion distributions; TCRα chains paired with cluster member TCRβ chains in the pairSEQ dataset of (*Howie et al., 2015*).
DOI: https://doi.org/10.7554/eLife.38358.009

manifests as occurrence pattern clustering. The other large, non-HLA associated TCR cluster had a number of distinctive features as well: strong preference for the TRBV06 family, followed by TRBV20 and TRBV04 (*Figure 3—figure supplement 2*); low numbers of inserted 'N' nucleotides; and a skewed age distribution biased toward younger subjects (*Figure 3—figure supplement 3*). These features, together with the lack of apparent HLA restriction, suggested that this cluster represented an invariant T cell subset, specifically MAIT (mucosal-associated invariant T) cells (*Kjer-Nielsen et al., 2012*; *Venturi et al., 2013*; *Pogorelyy et al., 2017*). Since MAIT cells are defined primarily by their alpha chain sequences, we searched in a recently published paired dataset (*Howie et al., 2015*) for partner chains of the clustered TCR$\beta$ chain sequences, and found a striking number that matched the MAIT consensus (TRAV1-2 paired with TRAJ20/TRAJ33 and a 12 residue CDR3, *Figure 3—figure supplement 3D*). We also looked for these clustered TCRs in a recently published MAIT cell sequence dataset (*Howson et al., 2018*) and found that 93 of the 138 cluster member TCRs occurred among the 31,654 unique TCRs from this dataset; of these 93 TCR$\beta$ chains, 27 were found among the 78 most commonly occurring TCRs in the dataset (the TCRs occurring in at least 7 of the 24 sequenced repertoires), a highly significant overlap ($P < 2 \times 10^{-52}$ in a one-sided hypergeometric test). These concordances indicate that our untargeted approach has detected a well-studied T cell subset de novo through analysis of occurrence patterns.

## HLA-associated TCRs

These analyses suggested to us that TCR co-occurrence patterns across the full cohort of subjects are strongly influenced by the distribution of the HLA alleles, in accordance with the expectation that the majority of $\alpha\beta$ TCRs are HLA-restricted. Covariation between TCRs responding to the same HLA-restricted epitopes would only be expected in subjects positive for the restricting alleles, with TCR presence and absence outside these subjects likely introducing noise into the co-occurrence analysis. We therefore decided to analyze patterns of TCR co-occurrence within subsets of the cohort positive for specific HLA alleles, and to restrict our co-occurrence analysis to TCRs having a statistically significant association with the specific allele defining the cohort subset. To begin, we performed a comprehensive analysis of TCR-HLA association.

At a false discovery rate of 0.05 (estimated from shuffling experiments; see Materials and methods), we were able to assign 16,951 TCR$\beta$ sequences to an HLA allele (or alleles: DQ and DP alleles were analyzed as $\alpha\beta$ pairs, and there were 5 DR/DQ haplotypes whose component alleles were so highly correlated across our cohort that we could not assign TCR associations to individual DR or DQ components; see Materials and methods). *Table 1* lists the top 50 HLA-associated TCR sequences by association $p$-value and top 10 associated TCRs for the well-studied A*02:01 allele.

We find that 8 of the top 10 A*02:01-associated TCRs have been previously reported and annotated as being responsive to viral epitopes, specifically influenza M1$_{58}$ and Epstein-Barr virus (EBV) BMLF1$_{280}$ (*Shugay et al., 2018*; *Tickotsky et al., 2017*). Moreover, each of these 8 TCR$\beta$ chains is present in a recent experimental dataset (*Dash et al., 2017*) that included tetramer-sorted TCRs positive for these two epitopes; each TCR has a clear similarity to one of the consensus epitope-specific repertoire clusters identified in that work, with the EBV TRBV20, TRBV29, and TRBV14 TCRs, respectively, matching the three largest branches of the BMLF1$_{280}$ TCR tree, and the three influenza M1$_{58}$ TCRs all matching the dominant TRBV19 'RS' motif consensus (*Figure 4—figure supplement 2*). TCRs with annotation matches are sparser in the top 50 across all other alleles, which is likely due in part to a paucity of experimentally characterized non-A*02 TCRs, however we again see EBV-epitope responsive TCRs (with B*08:01 and B*35:01 restriction).

A global comparison of TCR feature distributions for HLA-associated versus non-HLA-associated TCRs provides further evidence of functional selection. As shown in *Figure 4A*, HLA-associated TCRs are on average more clonally expanded than a set of background, non-HLA associated TCRs with matching frequencies in the cohort. They also have lower generation probabilities—are harder to make under a simple random model of the VDJ rearrangement process—which suggests that their observed cohort frequencies may be elevated by selection (*Figure 4B*, see Materials and methods for further details on the calculation of clonal expansion indices and generation probabilities; also see (*Pogorelyy et al., 2018*)). Examination of two-dimensional feature distributions suggests that these shifts are correlated, with HLA-associated TCRs showing an excess of lower-probability, clonally expanded TCRs (*Figure 4C*); this trend appears stronger for class-I associated TCRs than for class II-associated TCRs (*Figure 4—figure supplement 1*).

**Table 1.** The top 50 most significant HLA-associated public TCRβ chains and the top 10 for A*02:01 (indicated in bold).

| Association p-value | Overlap* | TCR Subjects † | HLA subjects‡ | Total subjects§ | V-family | CDR3 | HLA allele# | Epitope annotation |
|---|---|---|---|---|---|---|---|---|
| 3.7e-90 | 231 | 267 | 268 | 629 | TRBV19 | CASSIRSSYEQYF | **A*02:01** | Influenza virus |
| 2.4e-72 | 179 | 191 | 268 | 629 | TRBV29 | CSVGTGGTNEKLFF | **A*02:01** | Epstein-Barr virus |
| 3.8e-66 | 107 | 124 | 134 | 522 | TRBV20 | CSARNRDYGYTF | DRB1*03:01-DQ | |
| 1.9e-65 | 92 | 95 | 151 | 630 | TRBV05 | CASSLVVSPYEQYF | DRB1*07:01 | |
| 6.7e-64 | 91 | 94 | 134 | 522 | TRBV30 | CAWSRDSGSGNTIYF | DRB1*15:01-DQ | |
| 7.5e-59 | 51 | 53 | 66 | 630 | TRBV15 | CATSREEGDGYTF | B*35:01 | |
| 3.6e-57 | 89 | 96 | 134 | 522 | TRBV11 | CASSPGQGPGNTIYF | DRB1*15:01-DQ | |
| 7.4e-56 | 57 | 57 | 95 | 630 | TRBV02 | CASSENQGSQPQHF | DRB1*04:01 | |
| 1.5e-52 | 86 | 87 | 184 | 629 | TRBV06 | CASSYDSGTGELFF | C*07:01 | |
| 3.3e-52 | 136 | 143 | 268 | 629 | TRBV19 | CASSIRSAYEQYF | **A*02:01** | Influenza virus |
| 1.2e-51 | 71 | 96 | 94 | 630 | TRBV27 | CASSLGGQNYGYTF | B*44:02 | |
| 1.8e-50 | 52 | 52 | 94 | 630 | TRBV28 | CASSSSPLNYGYTF | DRB1*01:01 | |
| 3.8e-49 | 69 | 71 | 142 | 630 | TRBV04 | CASSPGQGEGYEQYF | B*08:01 | Epstein-Barr virus |
| 6.3e-49 | 92 | 98 | 189 | 629 | TRBV11 | CASSFGQMNTEAFF | A*01:01 | |
| 1.3e-48 | 73 | 75 | 156 | 630 | TRBV18 | CASSPPTESYGYTF | B*07:02 | |
| 3.2e-48 | 79 | 87 | 151 | 630 | TRBV14 | CASSQAGMNTEAFF | DRB1*07:01 | |
| 8.7e-47 | 49 | 49 | 95 | 630 | TRBV11 | CASSLDQGGSSSYNEQFF | DRB1*04:01 | |
| 3.2e-46 | 50 | 51 | 95 | 630 | TRBV20 | CSAQREYNEQFF | DRB1*04:01 | |
| 3.3e-46 | 68 | 69 | 134 | 522 | TRBV05 | CASSFWGRDTQYF | DRB1*03:01-DQ | |
| 3.3e-46 | 54 | 59 | 94 | 630 | TRBV05 | CASSWTGGGGANVLTF | DRB1*01:01 | |
| 3.1e-45 | 54 | 60 | 94 | 630 | TRBV02 | CASSEARGAGQPQHF | DRB1*01:01 | |
| 1.4e-44 | 41 | 42 | 69 | 630 | TRBV14 | CASSPLGPGNTIYF | DRB1*11:01 | |
| 2.4e-43 | 92 | 121 | 134 | 522 | TRBV07 | CASSPTGLQETQYF | DRB1*03:01-DQ | |
| 4.1e-43 | 43 | 52 | 61 | 630 | TRBV19 | CASSPTGGIYEQYF | B*44:03 | Multiple sclerosis |
| 4.5e-43 | 39 | 40 | 66 | 629 | TRBV10 | CASSESPGNSNQPQHF | C*12:03 | |
| 6.7e-43 | 76 | 86 | 134 | 522 | TRBV28 | CASRGRPEAFF | DRB1*15:01-DQ | |
| 7.5e-43 | 50 | 54 | 94 | 630 | TRBV19 | CASSPTQNTEAFF | DRB1*01:01 | |
| 1.7e-42 | 84 | 110 | 142 | 630 | TRBV07 | CASSSGPNYEQYF | B*08:01 | |
| 1.7e-42 | 61 | 81 | 95 | 630 | TRBV05 | CASSFPGEDTQYF | DRB1*04:01 | |
| 1.3e-41 | 47 | 49 | 95 | 630 | TRBV18 | CASSPPAGAAYEQYF | DRB1*04:01 | |
| 1.5e-41 | 75 | 87 | 151 | 630 | TRBV28 | CASSLTSGGQETQYF | DRB1*07:01 | |
| 2.3e-41 | 64 | 67 | 151 | 630 | TRBV07 | CASSLGQGFYNSPLHF | DRB1*07:01 | |
| 8.2e-40 | 77 | 92 | 134 | 522 | TRBV19 | CASSISVYGYTF | DRB1*15:01-DQ | |
| 2.4e-39 | 43 | 54 | 66 | 630 | TRBV10 | CAISTGDSNQPQHF | B*35:01 | Epstein-Barr virus |
| 3.4e-39 | 115 | 193 | 156 | 630 | TRBV09 | CASSGNEQFF | B*07:02 | |
| 9.5e-39 | 151 | 260 | 189 | 629 | TRBV19 | CASSIRDSNQPQHF | A*01:01 | |
| 1.2e-38 | 100 | 103 | 268 | 629 | TRBV20 | CSARDGTGNGYTF | **A*02:01** | Epstein-Barr virus |
| 1.3e-38 | 56 | 60 | 130 | 629 | TRBV25 | CASSEYSLTDTQYF | C*04:01 | |
| 2.1e-38 | 109 | 116 | 268 | 629 | TRBV20 | CSARDRTGNGYTF | **A*02:01** | Epstein-Barr virus |

*Table 1 continued on next page*

*Table 1 continued*

| Association *p*-value | Overlap* | TCR Subjects † | HLA subjects‡ | Total subjects§ | V-family | CDR3 | HLA allele# | Epitope annotation |
|---|---|---|---|---|---|---|---|---|
| 2.3e-38 | 102 | 106 | 268 | 629 | TRBV19 | CASSVRSSYEQYF | **A*02:01** | Influenza virus |
| 6.4e-38 | 54 | 54 | 151 | 630 | TRBV10 | CAISESQDLNTEAFF | DRB1*07:01 | |
| 1.1e-37 | 43 | 45 | 94 | 630 | TRBV07 | CASSLAGPPNSPLHF | DRB1*01:01 | |
| 1.2e-37 | 44 | 60 | 66 | 630 | TRBV09 | CASSARTGELFF | B*35:01 | Epstein-Barr virus |
| 3.3e-37 | 79 | 88 | 189 | 629 | TRBV19 | CASSIDGEETQYF | A*01:01 | |
| 5.4e-37 | 64 | 70 | 134 | 522 | TRBV05 | CASSLESPNYGYTF | DRB1*03:01-DQ | |
| 2.0e-36 | 38 | 43 | 69 | 630 | TRBV06 | CASGAGHTDTQYF | DRB1*11:01 | |
| 2.9e-36 | 54 | 55 | 151 | 630 | TRBV05 | CASSLVVQPYEQYF | DRB1*07:01 | |
| 3.3e-36 | 57 | 81 | 95 | 630 | TRBV11 | CASSPGQDYGYTF | DRB1*04:01 | |
| 2.4e-35 | 50 | 53 | 109 | 522 | TRBV27 | CASNRQGPNTEAFF | DQB1*03:01-DQA1*05:05 | |
| 5.7e-35 | 75 | 95 | 134 | 522 | TRBV18 | CASSGQANTEAFF | DRB1*03:01-DQ | |
| 2.2e-33 | 86 | 88 | 268 | 629 | TRBV14 | CASSQSPGGTQYF | **A*02:01** | Epstein-Barr virus |
| 1.8e-32 | 84 | 86 | 268 | 629 | TRBV10 | CASSEDGMNTEAFF | **A*02:01** | |
| 4.3e-32 | 86 | 89 | 268 | 629 | TRBV05 | CASSLEGQASSYEQYF | **A*02:01** | Melanoma |
| 4.3e-32 | 86 | 89 | 268 | 629 | TRBV29 | CSVGSGGTNEKLFF | **A*02:01** | Epstein-Barr virus |

*Number of subjects positive for both the TCRβ chain and the indicated HLA allele.

†Number of subjects positive for the TCRβ chain with available HLA typing at the corresponding locus.

‡Number of subjects positive for the indicated HLA allele.

§Total number of subjects with available HLA typing at the corresponding locus.

#The following DR-DQ haplotype abbreviations are used: DRB1*03:01-DQ (DRB1*03:01-DQA1*05:01-DQB1*02:01) and DRB1*15:01-DQ (DRB1*15:01-DQA1*01:02-DQB1*06:02).

DOI: https://doi.org/10.7554/eLife.38358.011

To give a global picture of TCR-HLA association, we counted the number of significant TCR associations found for each HLA allele in the dataset, and plotted this number against the number of subjects in the cohort with that allele (*Figure 5*). As expected, the more common HLA alleles have on average greater numbers of associated TCRs (since greater numbers of subjects permit the identification of more public TCRs, and the statistical significance assigned to an observed association of fixed strength grows as the number of subjects increases). What was somewhat more surprising is that the slope of the correlation between cohort frequency and number of associated TCRs varied dramatically among the HLA loci, with HLA-DRB1 alleles having the largest number of associated TCRs for a given allele frequency and HLA-C alleles having the smallest. The best-fit slope for the five DR/DQ haplotypes (12.2) was roughly the sum of the DR (7.99) and DQ (3.39) slopes, suggesting as expected that these haplotypes were capturing TCRs associated with both the DR and DQ component alleles. The smaller rate of TCR association observed at the HLA-C locus could be explained by a relatively lower level of cell surface expression of HLA-C alleles as well as their greater tendency to interact with killer cell immunoglobulin-like receptors (KIR) on natural killer (NK) cells (*Kaur et al., 2017*).

We assessed the accuracy of our TCR:HLA associations in two ways. First, we compared our HLA allele assignments to those given in the VDJdb database (which provides the peptide:MHC target and hence a putative HLA restriction for all entries; [*Shugay et al., 2018*]) and found that 90% of the VDJdb assignments for TCRβ chains present in both sets matched our associations. This agreement increases to 96% after filtering for the highest level of supporting evidence (VDJdb score of 3). Interestingly, two of the mismatches with VDJdb score three were from the protein structural database: the allo-complex between the B*08-restricted LC13 TCR and HLA-B*44:05 (*Macdonald et al., 2009*), and the structure of the A*02-restricted JM22 TCR bridged to a class II allele by a

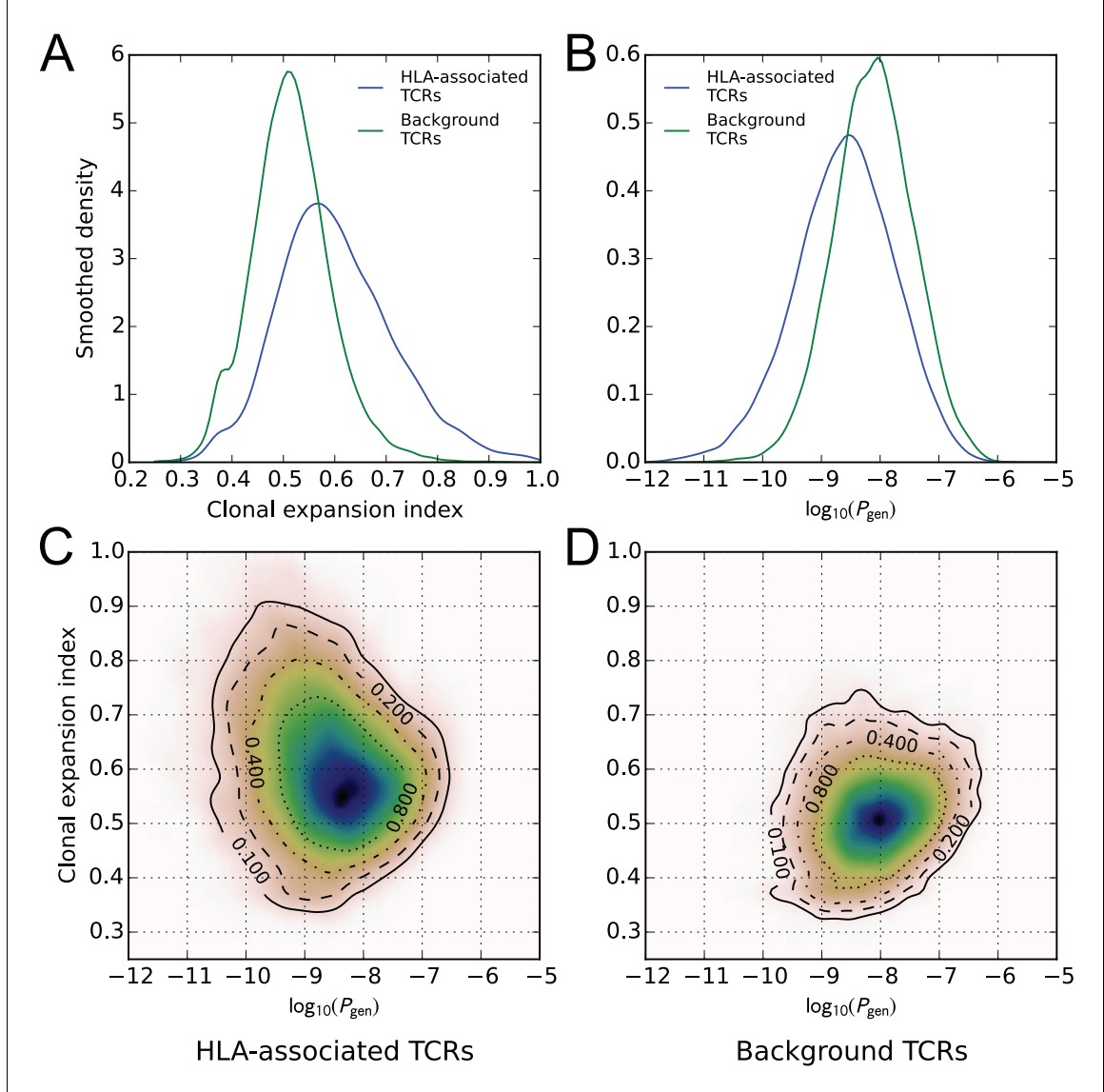

**Figure 4.** HLA-associated TCRs are more clonally expanded and have lower generation probabilities than equally common, non-HLA associated TCRs. (A) Comparison of clonal expansion index distributions for the set of HLA-associated TCRs (blue) and a cohort-frequency matched set of non HLA-associated TCRs (green). (B) Comparison of VDJ-rearrangement TCR generation probability ($P_{gen}$) distributions for the set of HLA-associated TCRs (blue) and a cohort-frequency matched set of non HLA-associated TCRs (green). (C) Two-dimensional probability density function (PDF) for the distribution of $P_{gen}$ versus clonal expansion index for HLA-associated TCRs. Contours indicate level sets of the PDF. (D) Two-dimensional probability density function (PDF) for the distribution of $P_{gen}$ versus clonal expansion index for background (non HLA-associated) TCRs whose cohort frequencies match the TCRs in (C).

DOI: https://doi.org/10.7554/eLife.38358.012

The following source data and figure supplements are available for figure 4:

**Source data 1.** Generation probabilities, clonal expansion indices, and allele associations for the TCRs analyzed here.
DOI: https://doi.org/10.7554/eLife.38358.015

**Figure supplement 1.** Two-dimensional feature distributions for HLA-associated TCR subsets defined by HLA locus.
DOI: https://doi.org/10.7554/eLife.38358.013

**Figure supplement 2.** TCRdist trees of experimentally determined pathogen-responsive TCRβ chains for two immunodominant epitopes, EBV BMLF1$_{280}$ and influenza M1$_{58}$, for comparison with TCRβ chains listed in *Table 1*.
DOI: https://doi.org/10.7554/eLife.38358.014

staphylococcal superantigen (*Saline et al., 2010*). In both of these cases, our data predict the canonical association: B*08 for the LC13 TCR$\beta$ chain and A*02 for the JM22 TCR$\beta$ chain. Second, we looked for HLA-associated public TCR$\beta$ chains in sequenced repertoires from T cell populations that were sorted for the presence of CD4/CD8 surface markers. One would expect that TCR$\beta$ chains associated with class I MHC molecules should be preferentially found in CD8+ populations, while class II-associated TCRs should be found in CD4+ populations. We selected four repertoire datasets (*Emerson et al., 2013*; *Rubelt et al., 2016*; *Li et al., 2016*; *Oakes et al., 2017*) with matched CD4 + and CD8+ repertoires from a total of 63 individuals, and we analyzed the occurrence patterns of our HLA-associated TCR$\beta$ chains in these sequence datasets, producing for each TCR$\beta$ counts of the number of CD4+ and CD8+ repertoires it was observed in. *Figure 5—figure supplement 1* shows that if we assign each TCR$\beta$ to the class (CD4+ or CD8+) with the higher count, these assignments are largely concordant with the MHC class of its associated HLA allele, and moreover this agreement increases as we increase either the stringency of HLA association or the stringency of the CD4/CD8

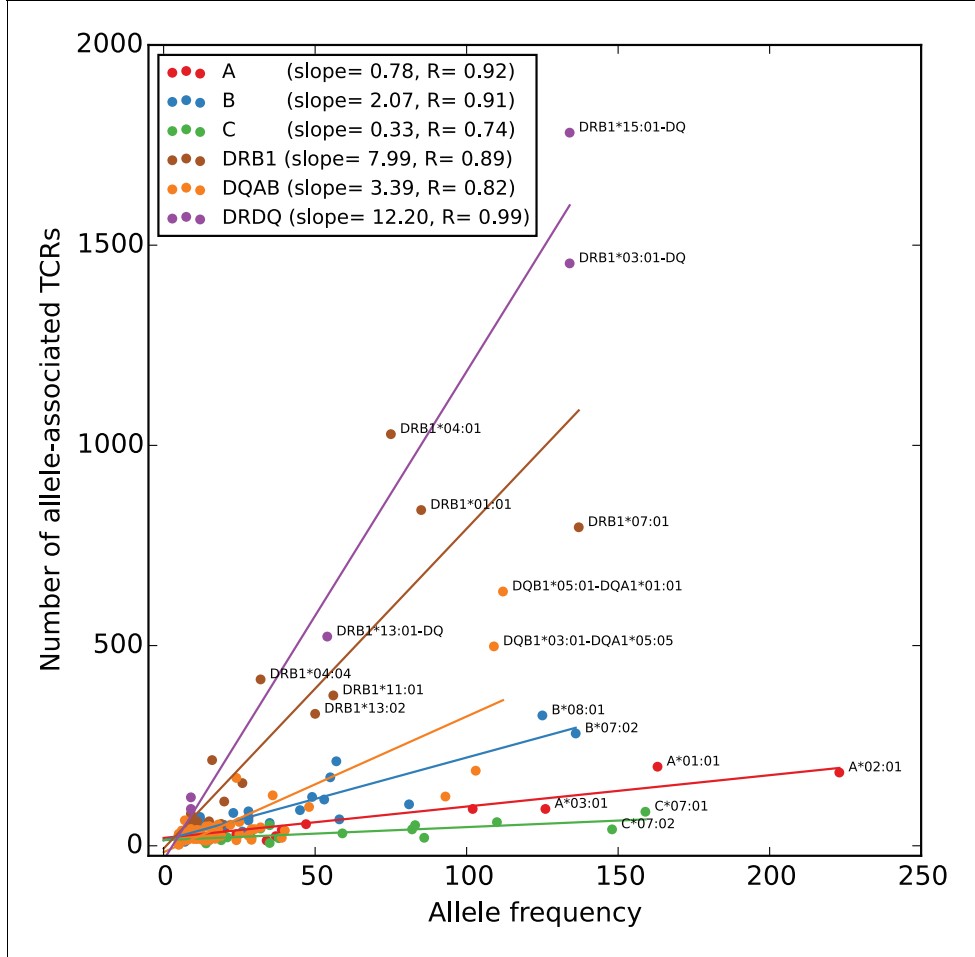

**Figure 5.** Rates of TCR association vary substantially across HLA loci. The number of HLA-associated TCRs (*y*-axis) is plotted as a function of allele frequency in the cohort (*x*-axis). Best fit lines are shown for each locus and also for the set of five DR/DQ haplotypes ('DRDQ') which could not be separated into component alleles in this cohort. The following DR-DQ haplotype abbreviations are used: DRB1*03:01-DQ (DRB1*03:01-DQA1*05:01-DQB1*02:01), DRB1*15:01-DQ (DRB1*15:01-DQA1*01:02-DQB1*06:02), and DRB1*13:01-DQ (DRB1*13:01-DQA1*01:03-DQB1*06:03).
DOI: https://doi.org/10.7554/eLife.38358.016

The following source data and figure supplement are available for figure 5:

**Source data 1.** Allele frequencies and numbers of associated TCRs.
DOI: https://doi.org/10.7554/eLife.38358.018

**Figure supplement 1.** HLA class associations are concordant with CD4/CD8 assignments based on independent repertoire data.
DOI: https://doi.org/10.7554/eLife.38358.017

assignment (i.e., the minimum absolute difference between the CD4 and CD8 repertoire counts; see Materials and methods).

## HLA-restricted TCR clusters

Having identified a set of HLA-associated TCR$\beta$ chains, we next sought to identify TCR clusters that might represent HLA-restricted responses to shared immune exposures. We performed this analysis for each HLA allele individually, restricting our clustering to the set of TCR chains significantly-associated with that allele and comparing occurrence patterns only over the subset of subjects positive for that allele. To reduce spurious co-occurrence signals driven by the presence/absence of other HLA alleles, we excluded TCR chains that were more strongly associated with a different HLA allele (i.e., not the one defining the cohort subset). The smaller size of many of these allele-positive cohort subsets reduces our statistical power to detect significant clusters using co-occurrence information. To counter this effect, we used the TCRdist similarity measure (*Dash et al., 2017*) to leverage the TCR sequence similarity which is often present within epitope-specific responses (*Dash et al., 2017*; *Glanville et al., 2017*) (see for example the A*02:01 TCRs in *Table 1* and *Figure 4—figure supplement 2*). We augmented the probabilistic similarity measure used to define neighbors for DBSCAN clustering to incorporate information about TCR sequence similarity (as measured by TCRdist), in addition to cohort co-occurrence (see Materials and methods). We independently clustered each allele's associated TCRs and merged the clustering results from all alleles; using the Holm multiple testing criterion (*Holm, 1979*) to limit the approximate family-wise error rate to 0.05, we found a total of 78 significant TCR clusters.

We analyzed the sequences and occurrence patterns of the TCRs belonging to these 78 clusters in order to assess their potential biological significance and prioritize them for further study (*Table 3*). Each cluster was assigned two scores (*Figure 6*): a size score ($S_{\text{size}}$, *x*-axis), reflecting the significance of seeing a cluster of that size given the total number of TCRs clustered for its associated allele, and a co-occurrence score ($Z_{\text{CO}}$, *y*-axis), reflecting the degree to which the TCRs in that cluster co-occur within its allele-positive cohort subset (see Materials and methods). In computing the co-occurrence score, we defined a subset of individuals with an apparent enrichment for the member TCRs in each cluster; the size of this enriched subset of subjects is given in the 'Subjects' column in *Table 3*. We rank ordered the 78 clusters based on the sum of their size and co-occurrence scores (weighted to equalize dynamic range); the top five clusters are presented in greater detail in *Figure 7* and *Figure 8*. HLA associations, member TCR and enriched subject counts, cluster center TCR sequences, scores, and annotations for all 78 clusters are given in *Table 3*.

We found that a surprising number of the most significant HLA-restricted clusters had links to common viral pathogens. For example, the top cluster by both size and co-occurrence (*Figure 7*, upper panels) is an A*24:02-associated group of highly similar TCR$\beta$ chains, five of which can be found in a set of 12 TCR$\beta$ sequences reported to respond to the parvovirus B19 epitope FYTPLADQF as part of a highly focused CD8+ response to acute B19 infection (*Kasprowicz et al., 2006*). The subject TCR-counts curve for this cluster (*Figure 7*, top right panel) shows a strong enrichment of member TCRs in roughly 30% of the A*24:02 repertoires, which is on the low end of prevalence estimates for this pathogen (*Heegaard and Brown, 2002*) and may suggest that, if cluster enrichment does correlate with B19 exposure, there are likely to be other genetic or epidemiologic factors that determine which B19-exposed individuals show enrichment. The second most significant cluster by both measures is an A*02:01-associated group of TRBV19 TCRs with a high frequency of matches to the influenza M1$_{58}$ response (41/43 TCRs, labeled 'INF-pGIL' for the first three letters of the GILGFVFTL epitope). Notably, the cluster member sequences recapitulate many of the core features of the tree of experimentally identified M1$_{58}$ TCRs (*Figure 4—figure supplement 2*): a dominant group of length 13 CDR3 sequences with an 'RS' sequence motif together with a smaller group of length 12 CDR3s with the consensus CASSIG.YGYTF.

Rounding out the top five, the third and fifth most significant clusters also appear to be pathogen-associated. Cluster #3 brings together a diverse set of DRB1*07:01-associated TCR$\beta$ chains (*Figure 8*, top dendrogram), none of which matched our annotation database. However, it was strongly associated with CMV serostatus: As is evident in the subject TCR-counts panel for this cluster (*Figure 8*, top right), there is a highly significant ($P<3 \times 10^{-19}$) association between CMV seropositivity (blue dots at the bottom of the panel) and cluster enrichment (here defined as a subject TCR count $\geq 3$). Finally, the B*08:01-associated cluster #5 (bottom panels in *Figure 8*) appears to be EBV-

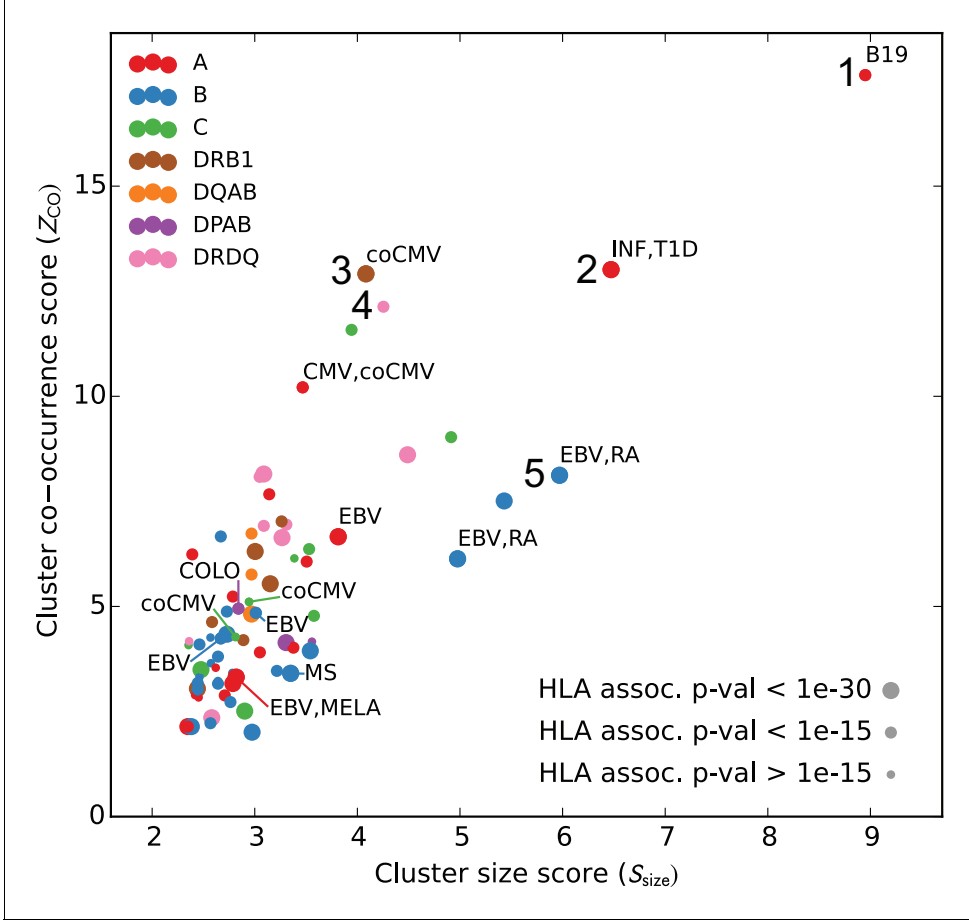

**Figure 6.** Many HLA-restricted TCR clusters contain TCR$\beta$ chains annotated as pathogen-responsive. Each point represents one of the 78 significant HLA-restricted TCR clusters, plotted based on a normalized cluster size score ($S_{size}$, $x$-axis) and an aggregate TCR co-occurrence score for the member TCRs ($Z_{CO}$, $y$-axis). Markers are colored by the locus of the restricting HLA allele and sized based on the strength of the association between cluster member TCRs and the HLA allele. The database annotations associated to TCRs in each cluster are summarized with text labels using the following abbreviations: B19 = parvovirus B19, INF = influenza, EBV = Epstein Barr Virus, RA = rheumatoid arthritis, MS = multiple sclerosis, MELA = melanoma, T1D = type one diabetes, CMV = cytomegalovirus. Clusters labeled 'coCMV' are significantly associated ($P<1 \times 10^{-5}$) with CMV seropositivity (see main text discussion of cluster #3). Clusters labeled 1–5 are discussed in the text and examined in greater detail in **Figure 7** and **Figure 8**.
DOI: https://doi.org/10.7554/eLife.38358.019

The following source data and figure supplement are available for figure 6:

**Source data 1.** Paired TCR$\alpha$ chain sequences from the pairSEQ dataset of (**Howie et al., 2015**) for all clusters with at least 2 matched TCR$\beta$ chains, along with a score for each cluster that assesses the degree of sequence similarity among the partner chains.
DOI: https://doi.org/10.7554/eLife.38358.021

**Figure supplement 1.** Distributions of cluster co-occurrence scores on the two validation cohorts.
DOI: https://doi.org/10.7554/eLife.38358.020

associated: four of the TCR$\beta$ chains in this cluster match TCRs annotated as binding to EBV epitopes (two matches for the B*08:01-restricted FLRGRAYGL epitope and two for the B*08:01-restricted RAKFKQLL epitope). The fact that this cluster brings together sequence-dissimilar TCRs that recognize different epitopes from the same pathogen supports the hypothesis that at least some of the observed co-occurrence may be driven by a shared exposure.

As a preliminary validation of the clusters identified here, we examined the occurrence patterns of cluster member TCRs in two independent cohorts: a set of 120 individuals ('Keck120') that formed the validation cohort for the original Emerson et al. study, and a set of 86 individuals ('Brit86') taken from the aging study of (**Britanova et al., 2016**). Whereas the Keck120 repertoires were generated using the same platform as our 666-member discovery cohort, the Brit86 repertoires were sequenced from cDNA libraries using 5'-template switching and unique molecular identifiers. In the

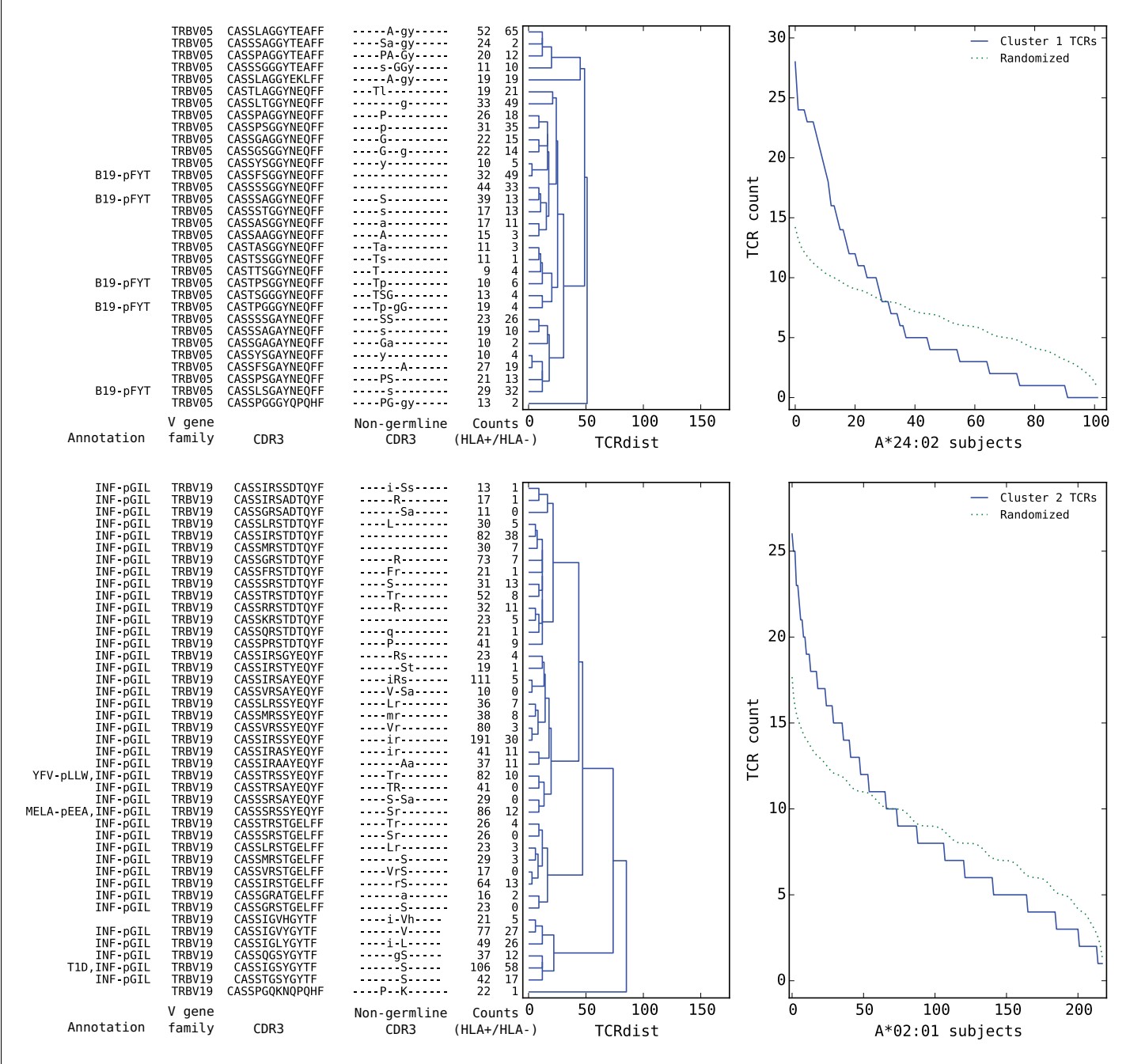

**Figure 7.** Top five HLA-restricted clusters (continued on following page). Details on the TCR sequences, occurrence patterns, and annotations for the five most significant clusters (labeled 1–5 in *Figure 6*) based on size and TCR co-occurrence scores. Each panel consists of a TCRdist dendrogram (left side, labeled with annotation, CDR3 sequence, and occurrence counts for the member TCRs) and a per-subject TCR count profile (right side) showing the aggregate occurrence pattern of the member TCRs (blue curve) and a control pattern (green curve) produced by averaging occurrence counts from multiple independent randomizations of the subject set for each TCR. The numbers in the two 'Counts' columns represent the number of HLA+ (left) and HLA- (right) subjects whose repertoire contained the corresponding TCR, where HLA± means positive/negative for the restricting allele (for example, A*24:02 in the case of cluster 1). Annotations use the following abbreviations: B19 (parvovirus B19), INF (influenza virus), YFV (yellow fever virus), MELA (melanoma), T1D (type 1 diabetes), EBV (Epstein-Barr virus), RA (rheumatoid arthritis). In cases where the peptide epitope for the annotation match is known, the first three peptide amino acids are given after '-p'. Non-germline CDR3 amino acids with 2 or 3 non-templated nucleotides in their codon are shown in uppercase, while amino acids with only a single non-templated coding nucleotide are shown in lowercase.
DOI: https://doi.org/10.7554/eLife.38358.022

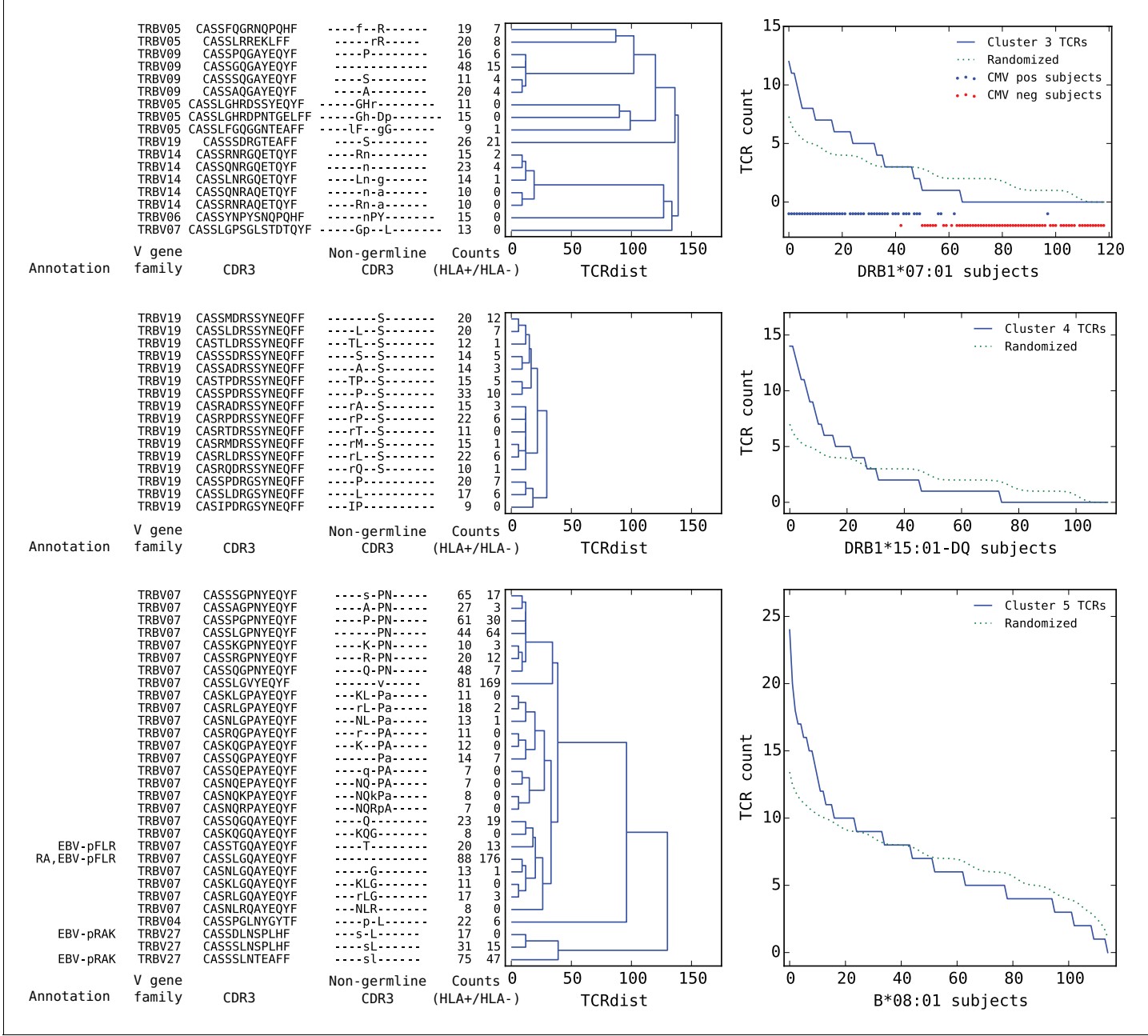

**Figure 8.** Top five HLA-restricted clusters (continued from previous page). Clusters 3–5; see preceding legend for details.

DOI: https://doi.org/10.7554/eLife.38358.023

absence of HLA typing information for these subjects, we simply evaluated the degree to which each cluster's member TCRs co-occurred over the entirety of each of these validation cohorts, using the co-occurrence score described above ($Z_{CO}^{Keck120}$ and $Z_{CO}^{Brit86}$ columns in *Table 3*). Although rare alleles and cluster-associated exposures may not occur with sufficient frequency in these smaller cohorts to generate co-occurrence signal, co-occurrence scores support the validity of the clusterings identified on the discovery cohort: 94% of the Keck120 scores and 92% of the Brit86 scores are greater than 0, indicating a tendency of the clustered TCRs to co-occur (smoothed score distributions are shown in *Figure 6—figure supplement 1*).

## Covariation between CDR3 sequence and HLA allele

Given our large dataset of HLA-associated TCR$\beta$ sequences, we set out to look for correlations between CDR3 sequence and HLA allele sequence. Previous studies have identified correlations between TCR V-gene usage and HLA alleles (*Sharon et al., 2016*; *Blevins et al., 2016*); these correlations are consistent with a picture of TCR:peptide:MHC interactions in which the CDR1 and CDR2 loops (whose sequence is determined by the V gene) primarily contact the MHC while the CDR3 loops contact the peptide. To complement these studies and leverage our large set of HLA-associated sequences, we set out to look for correlations between the CDR3 sequence itself and the HLA allele. In our previous work on epitope-specific TCRs (*Dash et al., 2017*), we identified a significant negative correlation between CDR3 charge and peptide charge, suggesting a tendency toward preserving charge complementarity across the TCR:pMHC interface. Although the CDR3 loop primarily contacts the MHC-bound peptide, computational analysis of solved TCR:peptide:MHC structures in the Protein Data Bank (*Berman et al., 2000*) (see Materials and methods) identified a number of HLA sequence positions that are frequently contacted by CDR3 amino acids (*Table 2*). For each frequently-contacted HLA position with charge variability among alleles we computed the covariation between HLA allele charge at that position and average CDR3 charge for allele-associated TCRs. Since portions of the CDR3 sequence are contributed by the V- and J-gene germline sequences, and covariations are known to exist between HLA and V-gene usage, we also performed a covariation analysis restricting to 'non-germline' CDR3 sequence positions whose coding sequence is determined by at least one non-templated insertion base (based on the most parsimonious VDJ reconstruction; see Materials and methods). We found a significant negative correlation ($R = -0.47, P{<}4 \times 10^{-4}$ for the full CDR3 sequence; $R = -0.52, P{<}7 \times 10^{-5}$ for the non-germline CDR3 sequence) between CDR3 charge and the charge at position 70 of the class II beta chain (correcting these $p$-values for the fact that we considered 7 positions yields $2.3 \times 10^{-3}$ and $4.3 \times 10^{-4}$). We did not see a significant correlation for the frequently contacted position on the class II alpha chain, perhaps due to the lack of sequence variation at the DR$\alpha$ locus and/or the more limited number of DQ$\alpha$ and DP$\alpha$ alleles. None of the five class I positions showed significant correlations, which could be due to their lower contact frequencies, a smaller average number of associated TCRs (51 for class I versus 309 for class II), bias toward A*02 in the structural database, or noise introduced from multiple contacted positions varying simultaneously. Further analysis of the class II correlation suggested that it was driven largely by HLA-DRB1 alleles: position 70 correlations were $-0.56$ versus $-0.10$ for DR and DQ, respectively, over the full CDR3 and $-0.64$ vs $-0.38$ for the non-germline CDR3. *Figure 9* provides further detail on this DRB1-TCR charge anti-correlation, including a structural superposition showing the proximity of position 70 to the TCR$\beta$ CDR3 loop.

**Table 2.** Covariation between HLA allele charge and average CDR3 charge of HLA-associated TCRs for HLA positions frequently contacted by CDR3 amino acids in solved TCR:pMHC crystal structures.

| MHC Class | Position[*] | Contact frequency[†] | Full CDR3 | | Non-germline CDR3[‡] | | AAs[§] |
|---|---|---|---|---|---|---|---|
| | | | R-value | *p*-value | R-value | *p*-value | |
| II-$\beta$ | 70 | 1.48 | −0.47 | 3.3e-04 | −0.52 | 6.1e-05 | DEGQR |
| II-$\alpha$ | 64 | 1.09 | −0.15 | 0.33 | −0.07 | 0.64 | ART |
| I | 152 | 0.47 | 0.00 | 0.99 | −0.04 | 0.72 | AERTVW |
| I | 151 | 0.46 | 0.08 | 0.50 | 0.06 | 0.59 | HR |
| I | 69 | 0.26 | −0.13 | 0.28 | −0.14 | 0.24 | ART |
| I | 76 | 0.21 | −0.08 | 0.49 | −0.14 | 0.25 | AEV |
| I | 70 | 0.12 | 0.02 | 0.86 | 0.08 | 0.50 | HKNQS |

[*]Only positions whose charge varies across alleles are included.

[†]Total number of CDR3 residues contacted (using a sidechain heavyatom distance threshold of 4.5 Å) divided by number of structures analyzed.

[‡]CDR3 charge is calculated over amino acids with at least one non-germline coding nucleotide.

[§]Amino acids present at this HLA position.

DOI: https://doi.org/10.7554/eLife.38358.024

## CMV-associated TCR$\beta$ chains are largely HLA-restricted

We analyzed the HLA associations of strongly CMV-associated TCR$\beta$ chains to gain insight into their predictive power across genetically diverse individuals. Here we change perspective somewhat from earlier sections, in that we select TCRs based on their CMV association and then evaluate HLA association, rather than the other way around. In their original study, Emerson et al. identified a set of TCR$\beta$ chains that were enriched in CMV seropositive individuals and showed that by counting these CMV-associated TCR$\beta$ chains in a query repertoire they could successfully predict CMV serostatus both in cross-validation and on an independent test cohort. The success of this prediction strategy across a diverse cohort of individuals raises the intriguing question of whether these TCR$\beta$s are primarily HLA-restricted in their occurrence and in their association with CMV, or whether they span multiple HLA types. To shed light on this question we focused on a set of 68 CMV-associated TCR$\beta$ chains whose co-occurrence with CMV seropositivity was significant at a p-value threshold of 1.5e-5 (corresponding to an FDR of 0.05; see Materials and methods). For each CMV-associated TCR$\beta$ chain, we identified its most strongly associated HLA allele and compared the p-value of this association to the p-value of its association with CMV (*Figure 10A*). From this plot we can see that the majority of the CMV-associated chains do appear to be HLA-associated, having p-values that exceed the FDR 0.05 threshold for HLA association. The excess of highly significant HLA-association p-values for these CMV-associated TCR$\beta$s can be seen in *Figure 10B*, which compares the observed p-value

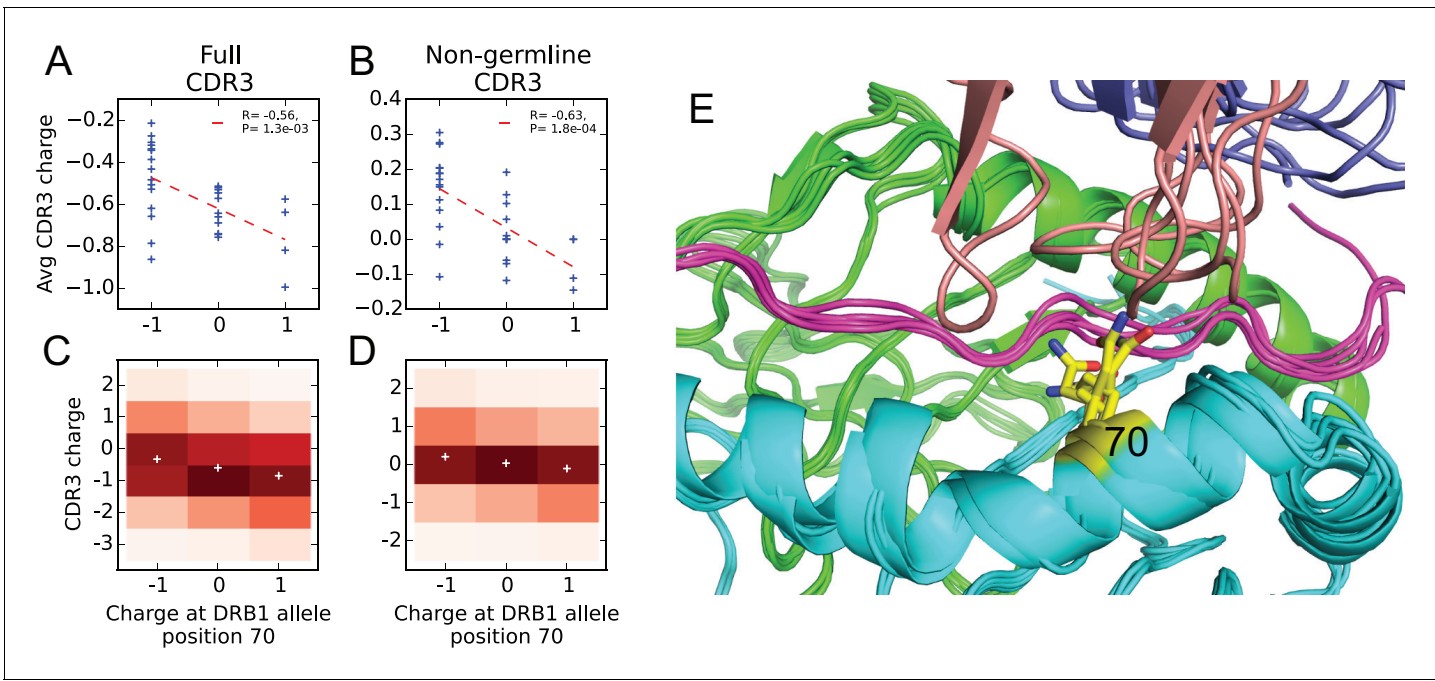

**Figure 9.** Negative correlation between HLA allele charge at DRB1 position 70 and CDR3 charge of HLA-associated TCRs. (A–B) Allele charge (x-axis) versus average CDR3 charge of allele-associated TCR$\beta$ chains (y-axis) for 30 HLA-DRB1 alleles. Charge of the CDR3 loop was calculated over the full CDR3 sequence (A) or over the subset of CDR3 amino acids with at least one non-germline coding nucleotide (B). Correlation p-values correspond to a 2-sided test of the null hypothesis that the slope is zero, as implemented in the function scipy.stats.linregress (N = 30 alleles). (C–D) CDR3 charge distributions for TCRs associated with alleles having defined charge at position 70 (x-axis) using the full (C) or non-germline (D) CDR3 sequence (mean values shown as white pluses). (E) Superposition of five TCR:peptide:HLA-DR crystal structures (PDB IDs 1j8h, 2iam, 2wbj, 3o6f, and 4e41; [*Hennecke and Wiley, 2002*; *Deng et al., 2007*; *Harkiolaki et al., 2009*; *Yin et al., 2011*; *Deng et al., 2012*]) showing the DR$\alpha$ chain in green, the DR$\beta$ chain in cyan, the peptide in magenta, the TCR$\beta$ chain in blue with the CDR3 loop colored reddish brown. The TCR$\alpha$ chain is omitted for clarity, and position 70 is highlighted in yellow.

DOI: https://doi.org/10.7554/eLife.38358.025

The following source data is available for figure 9:

**Source data 1.** Charge at position 70 and average CDR3 charge of allele-associated TCRs for 30 HLA-DRB1 alleles.
DOI: https://doi.org/10.7554/eLife.38358.026

distribution to a background distribution of HLA association $p$-values for randomly selected frequency-matched public TCR$\beta$s.

As a next step we looked to see whether these HLA associations fully explained the CMV association, in the sense that the CMV association was only present in subjects positive for the associated allele. For each of the 68 CMV-associated TCRs, we divided the cohort into subjects positive for its most strongly associated HLA allele and subjects negative for that allele. Here we considered both 2- and 4-digit resolution alleles when defining the most strongly associated allele, to allow for TCRs whose association extends beyond a single 4-digit allele. We computed association $p$-values between TCR occurrence and CMV seropositivity over these two cohort subsets independently and compared them (*Figure 10C*). We see that the majority of the points lie below the $y = x$ line—indicating a stronger CMV-association on the subset of the cohort positive for the associated allele—and also below the line corresponding to the expected minimum of 68 uniform random variables (i.e. the expected upper significance limit in the absence of CMV association on the allele-negative cohort subsets). There are however a few TCR$\beta$s which do not appear strongly HLA-associated and for which the CMV-association remains strong in the absence of their associated allele (the points above the line $y = x$ in *Figure 10C*). For example, the public TCR$\beta$ chain defined by TRBV07 and the CDR3 sequence CASSSDSGGTDTQYF (which corresponds to the highest point in *Figure 10C*) is strongly CMV-associated (22/23 subjects with this chain are CMV positive; $P < 3 \times 10^{-7}$) but does not show evidence of HLA association in our dataset. TCRs with HLA promiscuity may be especially interesting from a diagnostic perspective, since their phenotype associations may be more robust to differences in genetic background.

Finally, we looked to see whether CMV association completely explained the observed HLA associations, in the sense that a response to one or more CMV epitopes was likely the only driver of HLA association, or whether there might be evidence for other epitope-specific responses by these TCR$\beta$ chains or a more general affinity for the associated allele, perhaps driven by common self antigens. Put another way, do we see evidence for pre-existing enrichment of any of these TCR$\beta$ chains when their associated allele is present, even in the absence of CMV, which might suggest that the CMV response recruits from a pre-selected pool enriched for TCRs with intrinsic affinity for the restricting allele? To approach this question we split the cohort into CMV seropositive and seronegative subjects and computed, for each of the 68 CMV-associated TCRs, the strength of its association with its preferred allele over these two subsets separately. *Figure 10D* compares these HLA-association $p$-values computed over the subsets of the cohort positive (289 individuals, $x$-axis) and negative (352 individuals, $y$-axis) for CMV. We can see in this case that all of the associations on the CMV-positive subset are stronger than those on the CMV-negative subset, and indeed the CMV-negative $p$-values do not appear to exceed random expectation given the number of comparisons performed. Thus, the apparent lack of any significant HLA-association on the CMV-negative cohort subset suggests that the HLA associations of these CMV-predictive chains are largely driven by CMV exposure. A limitation of this analysis is that, although the CMV-negative subset of the cohort is larger than the CMV-positive subset, the number of TCR occurrences in the CMV-negative subset is likely lower than in the CMV-positive subset for these CMV-associated chains, which will limit the strength of the HLA associations that can be detected.

## Discussion

Each individual's repertoire of circulating immune receptors encodes information on their past and present exposures to infectious and autoimmune diseases, to antigenic stimuli in the environment, and to tumor-derived epitopes. Decoding this exposure information requires an ability to map from amino acid sequences of rearranged receptors to their eliciting antigens, either individually or collectively. One approach to developing such an antigen-mapping capability would involve collecting deep repertoire datasets and detailed phenotypic information on immune exposures for large cohorts of genetically diverse individuals. Correlation between immune exposure and receptor occurrence across such datasets could then be used to train statistical predictors of exposure, as demonstrated by Emerson et al. for CMV serostatus. The main difficulty with such an approach, beyond the cost of repertoire sequencing, is likely to be the challenge of assembling accurate and complete immune exposure information.

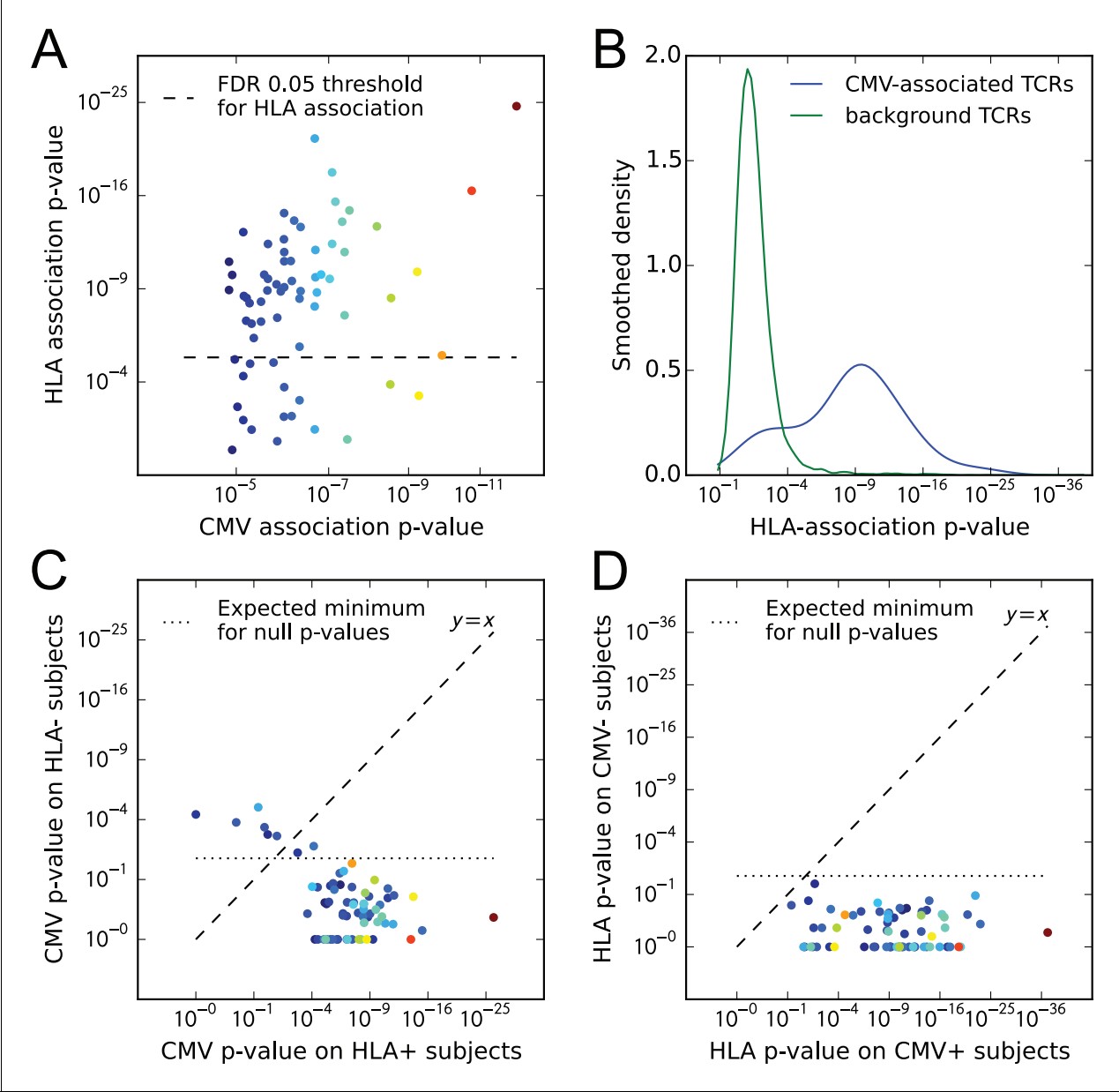

**Figure 10.** CMV-associated TCR$\beta$ chains are largely HLA-restricted. (**A**) Comparison of CMV-association (*x*-axis) and HLA-association (*y*-axis) *p*-values for 68 CMV-associated TCR$\beta$ chains shows that the majority are also HLA associated. (**B**) Smoothed densities comparing HLA-association *p*-value distributions for the 68 CMV-associated chains (blue) and a cohort-frequency matched set of 6800 randomly selected public TCR$\beta$ chains. CMV-associated TCRs are much more strongly HLA-associated than would be expected based solely on their cohort frequency. (**C**) CMV-association *p*-values computed over subsets of the cohort positive (*x*-axis) or negative (*y*-axis) for the HLA allele most strongly associated with each TCR. For most of the TCR chains, CMV association is restricted to the subset of the cohort positive for their associated HLA allele. (**D**) HLA-association *p*-values computed over CMV-positive (*x*-axis) or CMV-negative (*y*-axis) subsets of the cohort suggest that for these 68 CMV-associated TCR$\beta$ chains, HLA-association is driven solely by response to CMV (rather than generic affinity for their associated allele, for example, or additional self or viral epitopes). In panels (**A**), (**C**), and (**D**), points are colored by CMV-association *p*-value; in all panels we use a modified logarithmic scale based on the square root of the exponent when plotting *p*-values in order to avoid compression due to a few highly significant associations.

DOI: https://doi.org/10.7554/eLife.38358.027

The following source data is available for figure 10:

**Source data 1.** Full and subsetted CMV- and HLA- association *p*-values for 68 CMV-associated TCRs.

DOI: https://doi.org/10.7554/eLife.38358.028

For this reason, we set out to discover potential signatures of immune exposures de novo, in the absence of phenotypic information, using only the structure of the public repertoire—its receptor sequences and their occurrence patterns. By analyzing co-occurrence between pairs of public TCR$\beta$ chains and between individual TCR$\beta$ chains and HLA alleles, we were able to identify statistically significant clusters of co-occurring TCRs across a large cohort of individuals and in a variety of HLA backgrounds. Indirect evidence from sequence matches to experimentally-characterized receptors suggests that some of these TCR clusters may reflect hidden immune exposures shared among subsets of the cohort members; indeed, several of the most significant clusters appear linked to common viral pathogens (parvovirus B19, influenza, CMV, and EBV).

The results of this paper demonstrate the potential for a productive dialog between statistical analysis of TCR repertoires and immune exposure analysis. Specifically, sequences from the statistically-inferred clusters defined here could be tested for antigen reactivity or combined with immune exposure data to infer the driver of TCR expansion, as was done here for the handful of CMV-associated clusters based on CMV serostatus information. In either case our clustering approach will reduce the amount of independent data required, since the immune phenotype data is used for annotation of a modest number of defined TCR groupings rather than direct discovery of predictive TCRs from the entire public repertoire. We can also look for the presence of specific TCRs and TCR clusters identified here in other repertoire datasets, for example from studies of specific autoimmune diseases or pathogens, as a means of assigning putative functions. However the answer may not be entirely straightforward: it remains possible that enrichment for other cluster TCRs, rather than being associated with an exposure per se, is instead associated with some subject-specific genetic or epigenetic factor that determines whether a specific TCR response will be elicited by a given exposure.

The finding by Emerson et al.—now replicated and extended in this work—that there are large numbers of TCR$\beta$ chains whose occurrence patterns (independent of potential TCR$\alpha$ partners) are strongly associated with specific HLA alleles, raises the question of what selective forces drive these biased occurrence patterns. Our observations point to a potential role for responses to common pathogens in selecting some of these chains in an HLA-restricted manner. Self-antigens (presented in the thymus and/or the periphery) may also play a role in enriching for specific chains, as suggested by (*Madi et al., 2017*) in their work on TCR similarity networks formed by the most frequent CDR3 sequences. Our conclusions diverge somewhat from this previous work, which may be explained by the following factors: our use of HLA-association rather than intra-individual frequency as a filter for selecting TCRs, our inclusion of information on the V-gene family in addition to the CDR3 sequence when defining TCR sharing and computing TCR similarity, and our use of TCR occurrence patterns, rather than CDR3 edit distance, to discover TCR clusters. We also find it interesting that class II loci appear on average to have greater numbers of associated TCR$\beta$ chains than class I loci (*Figure 5*): presumably this reflects differences in selection and/or abundance between the CD4+ and CD8+ T cell compartments (*Sinclair et al., 2013*), but the underlying explanation for this trend is unclear, although a similar bias was observed by *Sharon et al., 2016*. One caveat is that it can be difficult to reliably assign TCR associations to individual members of groups of highly correlated HLA alleles; perfectly correlated alleles have been collapsed into haplotypes in our analysis, but there remain allele pairs (particularly between the HLA-DR and HLA-DQ loci) that strongly co-occur across the cohort. In addition, TCR$\beta$ chains associated with multiple HLA alleles (for example, because they recognize the same peptide presented by several different alleles) might be missed in our approach; although our analysis of HLA-association for CMV-associated TCR chains did not detect a substantial degree of HLA promiscuity, it remains to be seen whether this extends to other classes of functional TCRs. Alternative approaches that focus on other features, such as clonal abundance, to select TCR chains for clustering and downstream analysis are worth pursuing. It is also worth pointing out that our primary focus on presence/absence of TCR$\beta$ chains (rather than abundance) assumes relatively uniform sampling depths across the cohort; in the limit of very deep repertoire sequencing, pathogen-associated chains may be found (presumably in the naive pool) even in the absence of the associated immune challenge, while shallow sampling reliably picks out only the most expanded T cell clones. Here the use of clusters of responsive TCRs rather than individual chains lessens stochastic fluctuations in TCR occurrence patterns, providing some measure of robustness.

We look forward to the accumulation of new data sets, which will enable future researchers to move beyond the limitations of the study presented here. An ideal study would perform discovery on repertoire data from multiple large cohorts, rather than the single large cohort generated with a

single sequencing platform. Although we do validate TCR clusters on two independent datasets, with one from a different immune profiling technology, performing discovery on multiple large cohorts would presumably give more robust results. Future analyses of independent, HLA-typed cohorts will provide additional validation of trends seen here. The lack of sequenced TCR$\alpha$ or paired $\alpha/\beta$ repertoires for this cohort limits the features we can detect and may introduce bias into some of our conclusions. Certain T cell subsets, such as MAIT and invariant natural killer T cells, are more easily recognized from $\alpha$ chain sequence data. It is likely that many TCRs that are associated with specific immune exposures when considered as paired TCR chains are not detectably associated with those exposures (or with other TCRs responding to those exposures) when analyzing only the $\alpha$ or $\beta$ chain alone: indeed it is somewhat surprising that we find as many apparent associations and co-occurring clusters as we do given that we are considering only the TCR$\beta$ chain. Greater sequencing depth and/or analysis of sorted T cell populations will likely be required of future studies that aim to examine the impact of HLA on the composition of the naive T cell repertoire. We also hope that future studies will have rich immune exposure data beyond CMV serostatus: although the cohort members were all nominally healthy at the time of sampling, it is likely that there are a variety of immune exposures, some presaging future pathologies, that can be observed in a diverse collection of 650+ individuals. As an example, two of our EBV-annotated clusters contain TCR$\beta$ chains also seen in the context of rheumatoid arthritis: cross-reactivity between pathogen and autoimmune epitopes may mean that TCR clusters discovered on the basis of common infections also provide information relevant in the context of autoimmunity.

## Materials and methods

### Datasets

TCR$\beta$ repertoire sequence data for the 666 members of the discovery cohort was downloaded from the Adaptive biotechnologies website using the link provided in the original (*Emerson et al., 2017*) publication (https://clients.adaptivebiotech.com/pub/Emerson-2017-NatGen). The repertoire sequence data for the 120 individuals in the 'Keck120' validation set was included in the same download. Repertoire sequence data for the 86 individuals in the 'Brit86' validation set was downloaded from the NCBI SRA archive using the Bioproject accession PRJNA316572 (*Britanova et al., 2016*) and processed using scripts and data supplied by the authors (https://github.com/mikessh/aging-study) in order to demultiplex the samples and remove technical replicates. Repertoire sequence data for TCR$\beta$ chains from MAIT cells was downloaded from the NCBI SRA archive using the Bioproject accession PRJNA412739 (*Howson et al., 2018*). Repertoire sequence data for TCR$\beta$ chains from T cells sorted for CD4/CD8 surface markers were taken from the following studies: (*Emerson et al., 2013*), available for download at https://clients.adaptivebiotech.com/pub/emerson-2013-jim; (*Rubelt et al., 2016*), downloaded from the NCBI SRA archive using the Bioproject accession PRJNA300878; (*Li et al., 2016*), downloaded from the NCBI SRA archive using the Bioproject accession PRJNA348095; and (*Oakes et al., 2017*), downloaded from the NCBI SRA archive using the Bioproject accession PRJNA390125.

V and J genes were assigned by comparing the TCR nucleotide sequences to the IMGT/GENE-DB (*Giudicelli et al., 2005*) nucleotide sequences of the human TR genes (sequence data downloaded on 9/6/2017 from http://www.imgt.org/genedb/). CDR3 nucleotide and amino acid sequences and most-parsimonious VDJ recombination scenarios were assigned by the TCRdist pipeline (*Dash et al., 2017*) (the most parsimonious recombination scenario, used for identifying non-germline CDR3 amino acids, is the one requiring the fewest non-templated nucleotide insertions). To define the occurrence matrix of public TCRs and assess TCR-TCR, TCR-HLA and TCR-CMV association, a TCR$\beta$ chain was identified by its CDR3 amino acid sequence and its V-gene family (e.g., TRBV6-4*01 was reduced to TRBV06). TCR sequence reads for which a unique V-gene family could not be determined (due to equally well-matched V genes from different families, a rare occurrence in this dataset) were excluded from the analysis. The matrix $M$ of public TCR$\beta$ occurrences across the discovery cohort, HLA allele occurrence patterns, and other associated data needed to reproduce the findings of this study have been deposited in the Zenodo database (doi:10.5281/zenodo.1248193).

## Eliminating potential cross-contamination

A preliminary analysis of TCR sharing at the nucleotide level was conducted to identify potential cross-contamination in the discovery cohort repertoires. Each TCRβ nucleotide sequence that was found in multiple repertoires was assigned a generation probability ($P_{gen}$, see below) in order to identify nucleotide sequences with suspiciously high sharing rates among repertoires. Visual comparison of the sharing rate (the number of repertoires in which each TCRβ nucleotide sequence was found) to the generation probability (*Figure 11*) showed that the majority of highly-shared TCRs had correspondingly high generation probabilities; it also revealed a cluster of TCR chains with unexpectedly high sharing rates. Examination of the sequences of these highly-shared TCRs revealed them to be variants of the consensus sequence CFFKQKTAYEQYF (coding sequence: tgtttttttcaagcagaagacggcatacgagcagtacttc). Consultation with scientists at Adaptive Biotechnologies confirmed that these sequences were likely to represent a technical artifact of the sequencing pipeline. We elected to remove all TCRβ nucleotide sequences whose sharing rates put them outside the decision boundary indicated by the black line in *Figure 11*, which eliminated the vast majority of the artifactual variants as well as a handful of other highly shared, low-probability sequences (592 nucleotide sequences in total were removed).

## Measuring clonal expansion

Each public TCRβ chain was assigned a clonal expansion index ($I_{exp}$) determined by its frequencies in the repertoires in which it was found. First, the unique TCRβ chains present in each repertoire were ordered based on their inferred nucleic acid template count (*Carlson et al., 2013*), and assigned a rank ranging from 0 (lowest template count) to $S - 1$ (highest template count), where $S$ is the total number of chains present in the repertoire. TCRs with the same template count were assigned the same tied rank equal to the midpoint of the tied group. In order to compare across repertoires, the ranks for each repertoire were then normalized by dividing by the number of unique sequences in the repertoire. The clonal expansion index for a given public TCR $t$ was taken to be its average normalized rank for the repertoires in which it occurred:

$$I_{exp}(t) = \frac{1}{m} \sum_{i=1}^{m} \frac{r_i}{S_i - 1},$$

where the sum is taken over the $m$ repertoires in which $t$ is found, $r_i$ is the template-count rank of TCR $t$ in repertoire $i$, and $S_i$ is the total size of repertoire $i$.

## HLA typing

HLA genotyping was performed and confirmed by molecular means, including sequence specific oligonucleotide probe typing (SSOP), Sanger sequencing (SBT) or next generation sequencing (NGS) (*Smith et al., 2014*). Independently, HLA alleles were imputed using data generated by high density single-nucleotide polymorphism arrays as previously described (*Martin et al., 2017*). Imputed alleles were compared with HLA typing data from SBT and NGS, and used to resolve ambiguous HLA codes generated by SSOP and provide a uniform set of four digit allele assignments. HLA typing data availability varied across loci as follows: HLA-A (629 subjects), HLA-B (630 subjects), HLA-C (629 subjects), HLA-DRB1 (630 subjects), HLA-DQA1 (522 subjects), HLA-DQB1 (630 subjects), HLA-DPA1 (606 subjects), and HLA-DPB1 (472 subjects). When calculating the association p-values between TCRβ chains and HLA alleles reported in *Table 1*, the cohort was restricted to the subset of subjects with available HLA typing at the relevant locus. For comparing TCR association rates across loci in *Figure 5*, associations were calculated over the cohort subset (522 subjects) with typing data at all compared loci (A, B, C, DRB1, DQA1, and DQB1) in order to avoid spurious differences in association strengths arising from differential data availability among the loci. Due to their very strong linkage on our cohort, five DR-DQ haplotypes were treated as single allele units for association calculations and clustering: DRB1*03:01-DQA1*05:01-DQB1*02:01, DRB1*15:01-DQA1*01:02-DQB1*06:02, DRB1*13:01-DQA1*01:03-DQB1*06:03, DRB1*10:01-DQA1*01:05-DQB1*05:01, and DRB1*09:01-DQA1*03:02-DQB1*03:03.

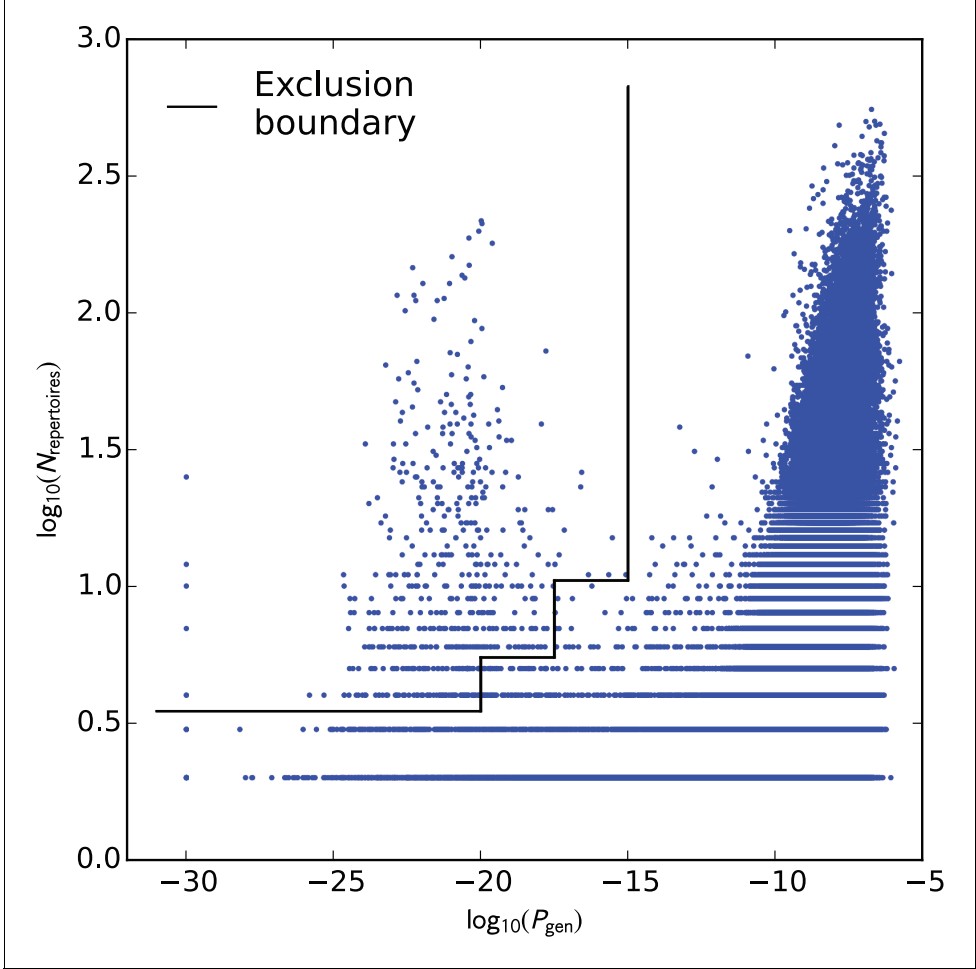

**Figure 11.** Analysis of TCR sharing at the nucleotide level and VDJ recombination probabilities helps to identify potential contamination. Each point represents a TCRβ nucleotide sequence that occurs in more than one repertoire, plotted according to its generation probability ($P_{gen}$, $x$-axis) and the number of repertoires in which it was seen ($N_{repertoires}$, $y$-axis). Very low probability nucleotide sequences that are shared across many repertoires represent potential cross-contamination, as confirmed for one large cluster of artifactual sequences (see the main text). We excluded all TCRβ nucleotide sequences lying above the boundary indicated by the black line ($N = 592$).
DOI: https://doi.org/10.7554/eLife.38358.029

The following source data is available for figure 11:

**Source data 1.** TCRβ nucleotide sequences excluded from our analysis.
DOI: https://doi.org/10.7554/eLife.38358.030

## TCR generation probability

We implemented a version of the probabilistic model proposed by Walczak and co-workers (*Murugan et al., 2012*) in order to assign to each public TCRβ chain (defined by a V-gene family and a CDR3 amino acid sequence) a generation probability, $P_{gen}$, which captures the probability of seeing that TCRβ in the preselection repertoire. $P_{gen}$ is calculated by summing the probabilities of the possible VDJ rearrangements that could have produced the observed TCR:

$$P_{gen}\left(V_{family}, CDR3_{aa}\right) = \sum_{s \in S} P(s)$$

where $S$ represents the set of possible VDJ recombination scenarios capable of producing the observed TCR V family and CDR3 amino acid sequence. To compute the probability of a given recombination scenario $s$, we use the factorization proposed by *Marcou et al. (2018)*, which

captures observed dependencies of V-, D-, and J-gene trimming on the identity of the trimmed gene and of inserted nucleotide identity on the identity of the preceding nucleotide:

$$
\begin{aligned}
P(s) = \ & P(V_s)P(D_s|J_s)P(J_s) \\
& \times P(\mathrm{del}_s V|V_s)P(\mathrm{del}_s D5', \mathrm{del}_s D3'|D_s)P(\mathrm{del}_s J|J_s) \\
& \times P(\mathrm{Ins}_s VD) \prod_i^{\mathrm{Ins}_s VD} P(n_i|n_{i-1}) \\
& \times P(\mathrm{Ins}_s DJ) \prod_i^{\mathrm{Ins}_s DJ} P(m_i|m_{i-1})
\end{aligned}
$$

Here the recombination scenario $s$ consists of a choice of V gene ($V_s$), D gene ($D_s$), J gene ($J_s$), number of nucleotides trimmed back from the end of the V gene ($\mathrm{del}_s V$) or J gene ($\mathrm{del}_s J$) or D gene ($\mathrm{del}_s D5'$ and $\mathrm{del}_s D3'$), number of nucleotides inserted between the V and D genes ($\mathrm{Ins}_s VD$) and between the D and J genes ($\mathrm{Ins}_s DJ$) and the identities of the inserted nucleotides ($\{n_i\}$ and $\{m_i\}$ respectively). At the start of the calculation, the CDR3 amino acid sequence is converted to a list of potential degenerate coding nucleotide sequences (here degenerate means that nucleotide class symbols such as W (for A and T) and R (for A and G) are allowed). Since each amino acid other than Leucine, Serine, and Arginine has a single degenerate codon (P=CCN, N = AAY, K = AAR, etc.) and these three amino acids have two such codons (S={TCN,AGY}, R={CGN,AGR}, L={CTN,TTR}), this list of nucleotide coding sequences is generally not too long. The generation probability is then taken to be the sum of the probabilities of these degenerate nucleotide sequences. Since the total number of possible recombination scenarios is in principle quite large, we make a number of approximations to speed the calculation: we limit *excess trimming* of genes to at most three nucleotides, where excess trimming is defined to be trimming back a germline gene nucleotide which matches the target CDR3 nucleotide (therefore requiring non-templated reinsertion of the same nucleotide); at most two palindromic nucleotides are allowed; sub-optimal D gene alignments are only considered up to a score gap of 2 matched nucleotides relative to the best match. The parameters of the probability model are fit by a simple iterative procedure in which we generate rearranged sequences using an initial model, compare the statistics of those sequences to statistics derived from observed out-of-frame rearrangements in the dataset, and adjust the probability model parameters to iteratively improve agreement. We compared the nucleotide sequence generation probabilities computed using our software with those computed using the published tool IGoR (*Marcou et al., 2018*) and found good overall agreement: a linear regression analysis of the $\log_{10}(P_{\mathrm{gen}})$ values gives a correlation coefficient $R = 0.97$ with slope of $0.98$ and an intercept of $0.22$ for a set of 800 randomly selected TCR$\beta$ chains.

## Co-occurrence calculations

We performed an analysis of covariation across the cohort for pairs of TCR chains and for TCR chains and HLA alleles (*Figure 12*). We used the hypergeometric distribution to assess the significance of an observed overlap between two subsets of the cohort (for example, the subset of subjects positive for a given HLA allele and the subset of subjects with a given TCR$\beta$ chain in their repertoires), taking our significance $p$-value to be the probability of seeing an equal or greater overlap if the two subsets had been chosen at random:

$$
P_{\mathrm{overlap}}(k, N_1, N_2, N) = \sum_{j \geq k} \frac{\binom{N_1}{j}\binom{N-N_1}{N_2-j}}{\binom{N}{N_2}}
$$

where $k$ is the size of the overlap, $N_1$ and $N_2$ are the sizes of the two subsets, and $N$ is the total cohort size (i.e., the number of individuals in the cohort). We use $P_{\mathrm{overlap}}$ to assess the significance of an overlap $C_a \cap C_t$ between an HLA allele $a$ found in the cohort subset $C_a$ and a TCR$\beta$ chain $t$ found in the cohort subset $C_t$ as follows:

$$
P_{\mathrm{HLA}}(a, t) = P_{\mathrm{overlap}}(|C_a \cap C_t|, |C_a|, |C_t|, N)
$$

where $|C|$ denotes the cardinality of the set $C$. A complication arises when assessing TCR-TCR co-occurrence in the presence of variable-sized repertoires: TCRs are more likely to come from the

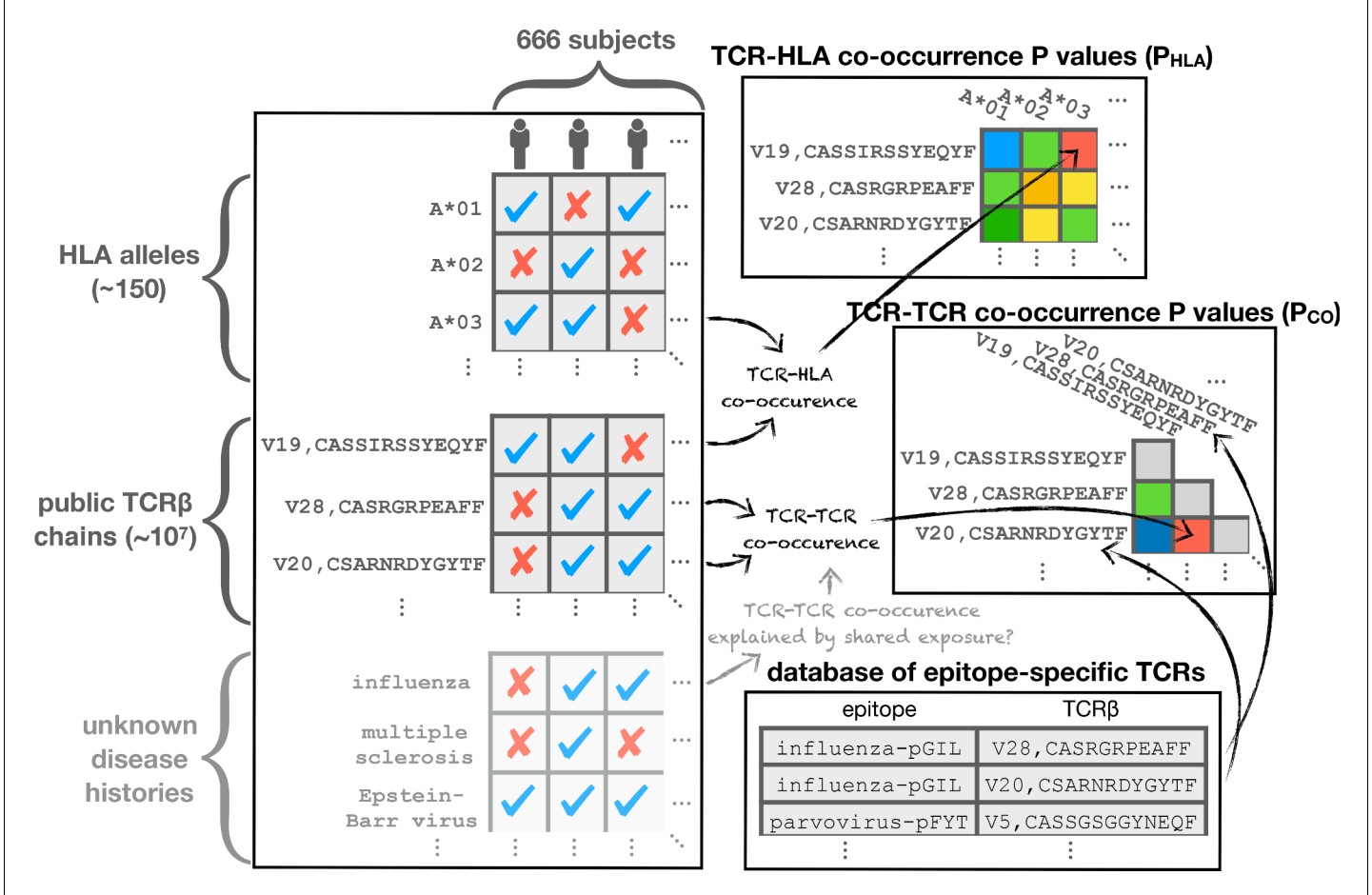

**Figure 12.** Schematic diagram illustrating the co-occurrence analysis. Co-occurrence *p*-values are calculated to assess TCR-TCR ($P_{CO}$) and TCR-HLA ($P_{HLA}$) covariation across the cohort. Shared response to unknown immune exposures may explain strongly co-occurring TCR pairs, while significant HLA association can highlight functional TCRs. TCR$\beta$ chains are compared to a set of previously characterized TCRs for annotation purposes.
DOI: https://doi.org/10.7554/eLife.38358.031

larger repertoires than the smaller ones, which violates the assumptions of the hypergeometric distribution and leads to inflated significance scores. In particular, when we use the hypergeometric distribution to model the overlap between the sets of subjects in which two TCR chains are found, we implicitly assume that all subjects are equally likely to belong to a TCR chain's subject set. If the subject repertoires vary in size, this assumption will not hold. For example, in the limit of a subject with an empty repertoire, no TCR subject sets will contain that subject, which will inflate all the overlap *p*-values since we are effectively overstating the size $N$ of the cohort by 1. On the other hand, if one of the subject repertoires contains all the public TCR chains, then each TCR-TCR overlap will automatically contain that subject, again inflating the *p*-values since we are artificially adding 1 to each of $k$, $N_1$, $N_2$, and $N$. We developed a simple heuristic to correct for this effect using a per-subject bias factor by defining

$$b_i = \frac{S_i N}{\sum_{j=1}^{N} S_j},$$

where $S_i$ is the size of repertoire $i$ and $N$ is the cohort size. To score an overlap between the occurrence patterns of two TCR$\beta$ chains $t$ and $t'$, where $t$ is found in the subset $C_t$ of the cohort, $t'$ is found in the subset $C_{t'}$, and their overlap $C_t \cap C_{t'}$ contains the $k$ subjects $s_1, ..., s_k$, we adjust the overlap *p*-value ($P_{overlap}$) by the product of the bias factors of the subjects in the overlap:

$$P_{\mathrm{CO}}(t,t') = \left(\prod_{j=1}^{k} b_{s_j}\right) P_{\mathrm{overlap}}(|C_t \cap C_{t'}|, |C_t|, |C_{t'}|, N)$$

Here we are multiplying the hypergeometric *p*-value ($P_{\mathrm{overlap}}$) by a term that corrects for the fact that not all overlaps of size $k$ are equally likely (the product of the $k$ bias factors captures the relative bias toward the observed overlap). This has the effect of decreasing the significance assigned to overlaps involving larger repertoires, yet remains fast to evaluate, an important consideration given that the all-vs-all TCR co-occurrence calculation involves about $10^{14}$ pairwise comparisons (and this calculation is repeated multiple times with shuffled occurrence patterns to estimate false-discovery rates). When clustering by co-occurrence, we augmented this heuristic *p*-value correction by also eliminating repertoires with very low (fewer than 30,000) or very high (more than 120,000) numbers of public TCR$\beta$ chains (nonzero entries in the occurrence matrix $M$), as well as five additional repertoires which showed anomalously high levels of TCR nucleotide sharing with another repertoire—all with the goal of reducing potential sources of spurious TCR-TCR co-occurrence signal.

## Estimating false-discovery rates

We used the approach of (*Storey and Tibshirani, 2003*) to estimate false-discovery rates for detecting associations between TCRs and HLA alleles and between TCRs and CMV seropositivity. Briefly, for a fixed significance threshold $P$ we estimate the false-discovery rate (FDR) by randomly permuting the HLA allele or CMV seropositivity assignments 20 times and computing the average number of significant associations discovered at the threshold $P$ in these shuffled datasets. The estimated FDR is then the ratio of this average shuffled association number to the number of significant associations discovered in the true dataset at the same threshold. In order to estimate a false-discovery rate for TCR-TCR co-occurrence over the full cohort, we performed 20 co-occurrence calculations on shuffled occurrence matrices, preserving the per-subject bias factors during shuffling by resampling each TCR's occurrence pattern with the bias distribution $\{b_i\}$ determined by the subject repertoire sizes.

## Assigning CD4+/CD8+ status to public TCRs

We assessed the accuracy of our TCR:HLA associations by looking for HLA-associated public TCR$\beta$ chains in sequenced repertoires from T cell populations that were sorted for the presence of CD4/CD8 surface markers. We selected four repertoire datasets with matched CD4+ and CD8+ repertoires from a total of 63 individuals (see the section Datasets for access details; [*Emerson et al., 2013*; *Rubelt et al., 2016*; *Li et al., 2016*; *Oakes et al., 2017*]). We analyzed the occurrence patterns of HLA-associated TCR$\beta$ chains in these sequence datasets, producing for each TCR$\beta$ counts of the number of CD4+ and CD8+ repertoires it was observed in ($N_{\mathrm{CD4}}$ and $N_{\mathrm{CD8}}$). TCR$\beta$ abundance levels within the individual repertoires were ignored; each occurrence in a repertoire contributed a single count to the respective CD4 or CD8 total (which therefore range between 0 and 63). Given a threshold $\delta$ on the CD4/CD8 counts difference, we assign to the CD4 compartment all TCRs for which $N_{\mathrm{CD4}} - N_{\mathrm{CD8}} \geq \delta$, and we assign to the CD8 compartment all TCRs for which $N_{\mathrm{CD8}} - N_{\mathrm{CD4}} \geq \delta$. *Figure 5—figure supplement 1* shows the concordance between these assignments and inferences based on the HLA class of the most strongly associated HLA allele, for all significantly associated TCR$\beta$ chains and for various thresholds $\delta$.

## TCR clustering

We used the DBSCAN (*Ester et al., 1996*) algorithm to cluster public TCR$\beta$ chains by their occurrence patterns. DBSCAN is a simple and robust clustering procedure that requires two input parameters: a similarity/distance threshold ($T_{\mathrm{sim}}$) at which two points in the dataset are considered to be neighbors, and a minimum number of neighbors ($N_{\mathrm{core}}$) for a point to be considered a *core*, as opposed to a *border*, point. DBSCAN clusters consist of the connected components of the neighbor-graph over the core points, together with any border point neighbors the core cluster members have. To prevent the discovery of fictitious clusters, $T_{\mathrm{sim}}$ and $N_{\mathrm{core}}$ can be selected so that core points (points with at least $N_{\mathrm{core}}$ neighbors) are unlikely to occur by chance. There is a trade-off between the two parameter settings: as $T_{\mathrm{sim}}$ is relaxed, points will tend to have more neighbors on average and thus $N_{\mathrm{core}}$ should be increased, which biases toward discovery of larger clusters; conversely,

more stringent settings of $T_{\text{sim}}$ are compatible with smaller values for $N_{\text{core}}$ which permits the discovery of smaller, more tightly linked clusters.

For clustering TCRs by co-occurrence over the full cohort, we used a threshold of $T_{\text{sim}} = 10^{-8}$ and chose a value for $N_{\text{core}}$ (6) such that no core points were found in any of the 20 shuffled datasets. In other words, two TCRs $t_1$ and $t_2$ were considered to be neighbors for DBSCAN clustering if $P_{\text{CO}}(t_1, t_2) < 10^{-8}$; a TCR was considered a core point if it had at least 6 neighbors. Choosing parameters for HLA-restricted TCR clustering was slightly more involved due to the variable number of clustered TCRs for different alleles, and the more complex nature of the similarity metric, whose dependence on TCR sequence makes shuffling-based approaches more challenging. To begin, we transformed the TCRdist sequence-similarity measure into a significance score $P_{\text{TCRdist}}$ which captures the probability of seeing an observed or smaller TCRdist score for two randomly selected TCR$\beta$ chains. Since public TCR$\beta$ chains are on average shorter and closer to germline than private TCRs, we derived the $P_{\text{TCRdist}}$ CDF by performing TCRdist calculations on randomly selected public TCRs seen in at least 5 repertoires. We identified neighbors for DBSCAN clustering using a similarity score $P_{\text{sim}}$ that combines co-occurrence and TCR sequence similarity:

$$P_{\text{sim}}(t_1, t_2) = f(P_{\text{TCRdist}}(t_1, t_2) \cdot P_{\text{CO}}(t_1, t_2))$$

where the transformation by $f(x) = x - x\log(x)$ corrects for taking the product of two $p$-values because $f(x)$ is the cumulative distribution function of the product of two uniform random variables. Thus, if $P_{\text{TCRdist}}$ and $P_{\text{CO}}$ are independent and uniformly distributed, the same will be true of $P_{\text{sim}}$.

For HLA-restricted clustering using this combined similarity measure we set a fixed value of $T_{\text{sim}} = 10^{-4}$ and adjusted the $N_{\text{core}}$ parameter as a function of the total number of TCRs clustered for each allele. As in global clustering, our goal was to choose $N_{\text{core}}$ such that core points were unlikely to occur by chance (more precisely, had a per-allele probability less than 0.05). We estimated the probability of seeing core points by modeling neighbor number using the binomial distribution, assuming that the observed neighbor number of a given TCR during clustering is determined by $M - 1$ independent Bernoulli-distributed neighborness tests with rate $r$, where $M$ is the number of clustered TCRs. Rather than assuming a fixed neighbor-rate $r$ across TCRs, we captured the observed variability in neighbor-rate (due, for example, to unequal V-gene frequencies and variable CDR3 lengths) by using a mixture of 20 rates $\{r_j\}$ estimated from similarity comparisons on randomly chosen public TCRs. More precisely, we choose the smallest value of $N_{\text{core}}$ for which the following inequality holds (where $M$ is the number of clustered TCRs for the allele in question):

$$\frac{M}{20} \sum_{j=1}^{20} \sum_{i=N_{\text{core}}}^{M-1} \binom{M-1}{i} r_j^i (1 - r_j)^{M-1-i} < 0.05$$

We also used this neighbor-number model to assign a $p$-value ($P_{\text{size}}$) to each cluster reflecting the likelihood of seeing the observed degree of clustering by chance. Since DBSCAN clusters are effectively single-linkage-style partitionings of the core points (together with any neighboring border points), they can have a variety of shapes, ranging from densely interconnected graphs, to extended clusters held together by local neighbor relationships (*Ester et al., 1996*). Modeling the total size of these arbitrary groupings is challenging, so we took the simpler and more conservative approach of assigning $p$-values based on the size of the largest TCR neighborhood (set of neighbors for a single TCR) contained within each cluster. We identified the member TCR with the greatest number of neighbors in each cluster (the *cluster center*) and computed the likelihood of seeing an equal or greater neighbor-number under the mixture model described above. This significance estimate is conservative in that it neglects clustering contributions from TCRs outside the neighborhood of the cluster center, however in practice we observed that the majority of TCR clusters were dominated by a single dense region of repertoire space and therefore reasonably well-captured by a single neighborhood. To control false discovery when combining DBSCAN clusters from independent clustering runs for different HLA alleles, we used the Holm method (*Holm, 1979*) applied to the sorted list of cluster $P_{\text{size}}$ values, with a target family-wise error rate (FWER) of 0.05 (i.e., we attempted to limit the overall probability of seeing a false cluster to 0.05). In the Holm FWER calculation we set the total number of hypotheses equal to the total number of TCRs clustered across all alleles minus the cumulative neighbor-count of the cluster centers (we exclude cluster center neighbors since their

neighbor counts are not independent of the neighbor count of the cluster center). When performing HLA-restricted clustering, each TCR$\beta$ chain was assigned to its most strongly associated HLA allele. Where two alleles had identical or nearly identical (within a factor of 1.25) association $p$-values, the TCR chain was included in the clustering analysis for both alleles.

## Analyzing TCR clusters

For each (global or HLA-restricted) TCR cluster, we analyzed the occurrence patterns of the member TCRs in order to identify a subset of the (full or allele-positive) cohort enriched for those TCRs. We counted the number of cluster member TCRs found in each subject's repertoire and sorted the subjects by this TCR count (rank plots in *Figure 3B–C* and in the right panels of *Figure 7*). For comparison, we generated control TCR count plots by independently resampling the subjects for each member TCR, preserving the frequency of each TCR and biasing by subject repertoire size. Each complete resampling of the cluster member TCR occurrence patterns produced a subject TCR rank plot; we repeated this resampling process 1000 times and averaged the rank plots to yield the green ('randomized') curves in *Figure 3B–C* and *Figure 7*. To compare the observed and randomized curves, we took a signed difference

$$D_{\mathrm{CO}} = \max_{1 \leq i \leq N} \left( \sum_{j \leq i} (C_j - R_j) + \sum_{j > i} (R_j - C_j) \right)$$

between the observed counts $C_j$ and the randomized counts $R_j$, where the value of the subject index $i = i_{\max}$ that maximizes the right-hand side in the equation above represents a switchpoint below which the observed counts generally exceed the randomized counts and above which the reverse is true (both sets of counts are sorted in decreasing order). We take this switchpoint $i_{\max}$ as an estimate of the number of enriched subjects for the given cluster (this is the value given in the 'Subjects' column in *Table 3*).

Since the raw $D_{\mathrm{CO}}$ values are not comparable between clusters of different sizes and for different alleles, we transformed these values to a Z-score ($Z_{\mathrm{CO}}$) by generating, for each cluster, 1000 additional random TCR count curves and computing the mean ($\mu_D$) and standard deviation ($\sigma_D$) of their $D_{\mathrm{CO}}^{\mathrm{rand}}$ score distribution:

$$Z_{\mathrm{CO}} = \frac{D_{\mathrm{CO}} - \mu_D}{\sigma_D}$$

We used this co-occurrence score $Z_{\mathrm{CO}}$ together with a log-transformed version of the cluster size $p$-value,

$$S_{\mathrm{size}} = \sqrt{-\log_{10}(P_{\mathrm{size}})}$$

for visualizing clustering results in *Figure 6* ($S_{\mathrm{size}}$ on the $x$-axis and $Z_{\mathrm{CO}}$ on the $y$-axis) and prioritizing individual clusters for detailed follow-up.

## TCR annotations

We annotated public TCRs in our dataset by matching their sequences against two publicly available datasets: VDJdb (*Shugay et al., 2018*), a curated database of TCR sequences with known antigen specificities (downloaded on 3/29/18; about $17,000$ human TCR$\beta$ entries) and McPAS-TCR (*Tickotsky et al., 2017*), a curated database of pathogen-associated TCR sequences (downloaded on 3/29/18; about $9,000$ human TCR$\beta$ entries). VDJdb entries are associated with a specific MHC-presented epitope, whereas McPAS-TCR also includes sequences of TCRs isolated from diseased tissues whose epitope specificity is not defined. We added to this merged annotation database the sequences of structurally characterized TCRs of known specificity (see below), as well as literature-derived TCRs from a handful of primary studies (*Dash et al., 2017*; *Glanville et al., 2017*; *Song et al., 2017*; *Kasprowicz et al., 2006*). For matches between HLA-associated TCRs and database TCRs of known specificity, we filtered for agreement (at 2-digit resolution) between the associated HLA allele in our dataset and the presenting allele from the database. In other words, TCRs belonging to B*08:01-restricted clusters were not annotated with matches to database TCRs that bind to A*02:01-presented peptides.

**Table 3.** HLA-restricted TCR clusters with size ($S_{\text{size}}$) and co-occurrence ($Z_{\text{CO}}$) scores, annotations (abbreviated as in *Figure 6*), and validation scores.

| Rank | HLA allele | Allele frequency | TCRs | Subjects | Cluster center | $S_{\text{size}}$ | $Z_{\text{CO}}$ | Annotations | $Z_{\text{CO}}^{\text{Keck120}}$ | $Z_{\text{CO}}^{\text{Brit86}}$ |
|---|---|---|---|---|---|---|---|---|---|---|
| 1 | A*24:02 | 102 | 32 | 29 | TRBV05,CASSGSGGYNEQFF | 8.95 | 17.64 | B19 | 10.38 | 6.74 |
| 2 | A*02:01 | 218 | 43 | 66 | TRBV19,CASSGRSTDTQYF | 6.47 | 13.01 | INF, T1D | 12.28 | 4.28 |
| 3 | DRB1*07:01 | 119 | 17 | 36 | TRBV09,CASSGQGAYEQYF | 4.08 | 12.91 | coCMV | 9.46 | 6.40 |
| 4 | DRB1*15:01-DQ | 112 | 16 | 27 | TRBV19,CASSPDRSSYNEQFF | 4.25 | 12.13 | | 1.65 | 1.72 |
| 5 | B*08:01 | 115 | 30 | 34 | TRBV07,CASSQGPAYEQYF | 5.97 | 8.12 | EBV, RA | 3.83 | 1.83 |
| 6 | C*04:01 | 104 | 7 | 24 | TRBV19,CASSPGGDYNEQFF | 3.94 | 11.58 | | 4.48 | 2.01 |
| 7 | C*04:01 | 104 | 11 | 20 | TRBV04,CASSHSGTGETYEQYF | 4.91 | 9.03 | | 7.52 | 1.66 |
| 8 | B*15:01 | 55 | 23 | 27 | TRBV19,CASSTTSGSYNEQFF | 5.43 | 7.51 | | 10.31 | 4.01 |
| 9 | DRB1*03:01-DQ | 108 | 26 | 39 | TRBV29,CSVAPGWGMNTEAFF | 4.49 | 8.61 | | 10.96 | 7.09 |
| 10 | A*01:01 | 154 | 8 | 44 | TRBV24,CATSDGDTQYF | 3.47 | 10.21 | CMV, coCMV | 3.80 | 2.42 |
| 11 | B*35:01 | 56 | 18 | 24 | TRBV10,CATGTGDSNQPQHF | 4.98 | 6.13 | EBV, RA | 4.50 | 5.42 |
| 12 | DRB1*03:01-DQ | 108 | 11 | 35 | TRBV07,CASSLSLAGSYNEQFF | 3.09 | 8.15 | | 5.35 | 1.40 |
| 13 | A*02:01 | 218 | 10 | 84 | TRBV20,CSARDRTGNGYTF | 3.81 | 6.66 | EBV | 7.14 | 3.50 |
| 14 | DRB1*15:01-DQ | 112 | 15 | 38 | TRBV05,CASSLRGVRTDTQYF | 3.05 | 8.08 | | 8.73 | 3.31 |
| 15 | A*01:01 | 154 | 6 | 30 | TRBV10,CAISESRASGDYNEQFF | 3.14 | 7.67 | | 11.31 | 2.99 |
| 16 | DRB1*13:01-DQ | 43 | 7 | 7 | TRBV20,CSASAGESNQPQHF | 3.14 | 7.64 | | −0.55 | −0.35 |
| 17 | DRB1*03:01-DQ | 108 | 16 | 32 | TRBV20,CSARGGGRSYEQYF | 3.31 | 6.95 | | 2.57 | 3.09 |
| 18 | DRB1*11:01 | 58 | 14 | 20 | TRBV06,CASSYSVRGRYSNQPQHF | 3.26 | 7.02 | | 8.72 | 3.44 |
| 19 | C*08:02 | 37 | 6 | 15 | TRBV28,CASSLGIHYEQYF | 3.53 | 6.37 | | 1.82 | 4.37 |
| 20 | DRB1*15:01-DQ | 112 | 13 | 51 | TRBV12,CASSLAGTEKLFF | 3.27 | 6.64 | | 4.61 | 3.01 |
| 21 | DRB1*03:01-DQ | 108 | 11 | 23 | TRBV05,CASSSTGLRSYEQYF | 3.09 | 6.92 | | 4.73 | 5.81 |
| 22 | A*02:01 | 218 | 7 | 64 | TRBV04,CASSQGTGRYEQYF | 3.51 | 6.07 | | 2.79 | 3.23 |
| 23 | C*03:04 | 72 | 5 | 13 | TRBV09,CASSVAYRGNEQFF | 3.39 | 6.14 | | 6.26 | 3.23 |
| 24 | DQB1*03:01-DQA1*05:05 | 84 | 10 | 39 | TRBV09,CASSVGTVQETQYF | 2.97 | 6.73 | | 3.02 | 3.54 |
| 25 | DRB1*04:01 | 78 | 25 | 35 | TRBV05,CASSRQGAGETQYF | 3.00 | 6.31 | | 5.82 | 1.55 |
| 26 | B*08:01 | 115 | 7 | 30 | TRBV12,CASSFEGLHGYTF | 2.67 | 6.67 | | 3.77 | 2.95 |
| 27 | C*04:01 | 104 | 6 | 25 | TRBV06,CASRTGLAGTDTQYF | 3.58 | 4.78 | | 3.53 | 3.76 |
| 28 | DRB1*07:01 | 119 | 9 | 42 | TRBV14,CASSLAGMNTEAFF | 3.15 | 5.54 | | 6.99 | 5.58 |
| 29 | DQB1*03:01-DQA1*05:05 | 84 | 7 | 36 | TRBV02,CASSELENTEAFF | 2.97 | 5.76 | | 5.25 | 3.24 |
| 30 | DPB1*03:01-DPA1*01:03 | 42 | 7 | 16 | TRBV30,CAWSADSNQPQHF | 3.56 | 4.16 | | 2.42 | 1.73 |
| 31 | B*15:01 | 55 | 18 | 27 | TRBV29,CSVETRDYEQYF | 3.54 | 3.94 | | 13.81 | 4.29 |
| 32 | A*01:01 | 154 | 4 | 26 | TRBV09,CASSVGVDSTDTQYF | 2.39 | 6.24 | | −0.31 | 2.17 |
| 33 | C*07:02 | 142 | 4 | 14 | TRBV25,CASSPGDEQYF | 2.94 | 5.11 | coCMV | 6.37 | 3.69 |
| 34 | B*08:01 | 115 | 6 | 38 | TRBV29,CSVGSGDYEQYF | 3.01 | 4.85 | EBV | 2.73 | 0.75 |
| 35 | A*01:01 | 154 | 6 | 37 | TRBV20,CSAPGQGAVEQYF | 2.79 | 5.24 | | 2.42 | 3.00 |
| 36 | A*23:01 | 22 | 5 | 7 | TRBV06,CASSDGNSGNTIYF | 3.38 | 4.02 | | 1.91 | 4.11 |
| 37 | DQB1*03:01-DQA1*05:05 | 84 | 7 | 29 | TRBV15,CATSRDPGGNQPQHF | 2.97 | 4.82 | | 5.00 | 2.67 |
| 38 | DPB1*04:01-DPA1*01:03 | 274 | 5 | 65 | TRBV19,CASSIKGDTEAFF | 3.31 | 4.14 | | 4.89 | 3.42 |
| 39 | DPB1*04:01-DPA1*01:03 | 274 | 4 | 55 | TRBV19,CASRLSGDTQYF | 2.84 | 4.95 | COLO | 3.80 | 1.25 |
| 40 | B*07:02 | 125 | 7 | 37 | TRBV02,CASRGETQYF | 2.73 | 4.88 | | 3.20 | 2.11 |
| 41 | B*44:03 | 41 | 9 | 20 | TRBV19,CASSATGGIYEQYF | 3.35 | 3.41 | MS | 6.61 | 8.76 |
| 42 | A*24:02 | 102 | 6 | 31 | TRBV30,CAWSPGTGDYEQYF | 3.05 | 3.91 | | 3.56 | 2.99 |
| 43 | DRB1*07:01 | 119 | 13 | 31 | TRBV18,CASSPSVRNTEAFF | 2.89 | 4.20 | | 5.32 | 0.96 |

*Table 3 continued on next page*

*Table 3 continued*

| Rank | HLA allele | Allele frequency | TCRs | Subjects | Cluster center | $S_{size}$ | $Z_{CO}$ | Annotations | $Z_{CO}^{Keck120}$ | $Z_{CO}^{Brit86}$ |
|---|---|---|---|---|---|---|---|---|---|---|
| 44 | B*57:01 | 27 | 5 | 14 | TRBV12,CASSPPEGETQYF | 3.22 | 3.47 | | 6.31 | 1.94 |
| 45 | C*06:02 | 74 | 4 | 14 | TRBV02,CASSAGTASTDTQYF | 2.81 | 4.27 | coCMV | 4.76 | 3.06 |
| 46 | A*11:01 | 47 | 5 | 7 | TRBV09,CASSPKGVGYEQYF | 2.75 | 4.31 | | 2.43 | 3.32 |
| 47 | DRB1*01:01 | 82 | 9 | 21 | TRBV19,CASSIPGLAYEQYF | 2.58 | 4.63 | | 0.96 | −0.49 |
| 48 | B*07:02 | 125 | 7 | 21 | TRBV09,CASSDRRGYTF | 2.73 | 4.34 | | 4.57 | 0.45 |
| 49 | B*08:01 | 115 | 6 | 22 | TRBV07,CASSSTGAGNQPQHF | 2.67 | 4.24 | EBV | 1.00 | 2.85 |
| 50 | B*18:01 | 46 | 5 | 6 | TRBV27,CASSPTSEDTQYF | 2.57 | 4.26 | | 5.79 | −0.23 |
| 51 | B*27:05 | 36 | 7 | 13 | TRBV06,CASSLRLAGLYEQYF | 2.64 | 3.81 | | 9.25 | 1.08 |
| 52 | B*35:01 | 56 | 4 | 7 | TRBV07,CASSQGPGRTYEQYF | 2.46 | 4.10 | | - | - |
| 53 | B*35:03 | 16 | 4 | 7 | TRBV10,CAISVGNEQFF | 2.78 | 3.42 | | 1.50 | 0.73 |
| 54 | A*02:01 | 218 | 5 | 126 | TRBV29,CSVGTGGTNEKLFF | 2.82 | 3.32 | EBV, MELA | 5.65 | 2.37 |
| 55 | DRB1*03:01-DQ | 108 | 6 | 18 | TRBV02,CASSAGAGTEAFF | 2.36 | 4.17 | | 0.98 | 2.79 |
| 56 | B*44:02 | 79 | 4 | 18 | TRBV02,CASSADSSYNEQFF | 2.57 | 3.65 | | 2.09 | 2.12 |
| 57 | C*03:04 | 72 | 3 | 8 | TRBV27,CASSPRPYNEQFF | 2.35 | 4.08 | | 1.36 | 3.22 |
| 58 | A*24:02 | 102 | 4 | 12 | TRBV20,CSAREDGHEQYF | 2.62 | 3.54 | | 0.83 | 2.94 |
| 59 | A*01:01 | 154 | 12 | 65 | TRBV19,CASSIRDHNQPQHF | 2.79 | 3.17 | | 8.44 | 2.33 |
| 60 | B*27:05 | 36 | 4 | 12 | TRBV07,CASSPPGGSAYNEQFF | 2.64 | 3.23 | | 1.13 | 2.12 |
| 61 | C*14:02 | 23 | 4 | 9 | TRBV02,CASSGDTSTNEKLFF | 2.48 | 3.50 | | 6.23 | - |
| 62 | B*27:05 | 36 | 9 | 12 | TRBV27,CASSSGTSGNNEQFF | 2.64 | 3.16 | | 4.32 | 3.24 |
| 63 | C*12:03 | 53 | 6 | 25 | TRBV15,CATSRENEKLFF | 2.90 | 2.51 | | 1.88 | 3.08 |
| 64 | A*68:01 | 29 | 4 | 16 | TRBV05,CASSLIATNEKLFF | 2.71 | 2.88 | | 3.67 | 1.23 |
| 65 | B*51:01 | 53 | 6 | 20 | TRBV04,CASSQDYPGGSYEQYF | 2.76 | 2.73 | | 6.43 | 5.18 |
| 66 | B*35:01 | 56 | 4 | 8 | TRBV27,CASSLGAATGELFF | 2.46 | 3.32 | | 4.52 | 3.01 |
| 67 | B*15:01 | 55 | 4 | 20 | TRBV06,CASSAGTGRYEQYF | 2.44 | 3.18 | | 2.40 | 2.23 |
| 68 | B*44:03 | 41 | 7 | 14 | TRBV07,CASSSGESGANVLTF | 2.97 | 2.01 | | 3.92 | 4.81 |
| 69 | DRB1*04:02 | 14 | 4 | 6 | TRBV03,CASSQASGGANEQFF | 2.44 | 3.04 | | 2.04 | 2.22 |
| 70 | B*15:01 | 55 | 4 | 10 | TRBV19,CASSHRGGNEQFF | 2.44 | 3.03 | | 0.92 | 3.58 |
| 71 | B*15:01 | 55 | 5 | 7 | TRBV05,CASSLGVSAGELFF | 2.44 | 2.98 | | −0.32 | −0.12 |
| 72 | A*32:01 | 34 | 3 | 5 | TRBV12,CASSYGPGNQPQHF | 2.45 | 2.84 | | 5.76 | 3.18 |
| 73 | A*02:01 | 218 | 4 | 23 | TRBV19,CASSTGTATNEKLFF | 2.42 | 2.89 | | 0.84 | - |
| 74 | DRB1*15:01-DQ | 112 | 7 | 51 | TRBV28,CASSLLGGQPQHF | 2.58 | 2.35 | | 0.66 | 1.89 |
| 75 | B*18:01 | 46 | 5 | 15 | TRBV27,CASSFPGKEQYF | 2.57 | 2.22 | | −0.35 | 5.62 |
| 76 | B*49:01 | 16 | 3 | 8 | TRBV29,CSVERGYNEQFF | 2.38 | 2.14 | | 1.03 | 0.43 |
| 77 | A*23:01 | 22 | 3 | 6 | TRBV20,CSARDREGAGYGYTF | 2.35 | 2.14 | | −0.16 | −0.12 |
| 78 | B*55:01 | 13 | 3 | 10 | TRBV19,CASRGGNQPQHF | 2.36 | 2.09 | | 0.95 | −0.28 |

DOI: https://doi.org/10.7554/eLife.38358.032

## Structural analysis

We analyzed a set of experimentally determined TCR:peptide-MHC structures to find MHC positions frequently contacted by the CDR3$\beta$ loop. Crystal structures of complexes involving human TCRs and human class I or class II HLA alleles (*Table 4*) were identified using BLAST (*Altschul et al., 1997*) searches against the RCSB PDB (*Berman et al., 2000*) sequence database (ftp://ftp.wwpdb.org/pub/pdb/derived_data/pdb_seqres.txt). Structural coverage of HLA loci and alleles is sparse and highly biased toward well studied alleles such as HLA-A*02. Given the high degree of structural similarity among class I and among class II MHC structures solved to date, we elected to share contact

**Table 4.** PDB structures analyzed.

| PDB ID* | HLA allele | Vα | Jα | CDR3α | Vβ | Jβ | CDR3β | Peptide |
|---|---|---|---|---|---|---|---|---|
| 5bs0 | A*01 | TRAV21*01 | TRAJ28*01 | CAVRPGGAGPFFVVF | TRBV5-1*01 | TRBJ2-7*01 | CASSFNMATGQYF | ESDPIVAQY |
| 3qdj | A*02 | TRAV12-2*01 | TRAJ23*01 | CAVNFGGGKLIF | TRBV6-4*01 | TRBJ1-1*01 | CASSLSFGTEAFF | AAGIGILTV |
| 4l3e | A*02 | TRAV12-2*01 | TRAJ23*01 | CAVNFGGGKLIF | TRBV6-4*01 | TRBJ1-1*01 | CASSWSFGTEAFF | ELAGIGILTV |
| 5e9d | A*02 | TRAV12-2*01 | TRAJ24*02 | CAVTKYSWGKLQF | TRBV6-5*01 | TRBJ2-7*01 | CASRPGWMAGGVELYF | ELAGIGILTV |
| 3qfj | A*02 | TRAV12-2*01 | TRAJ24*02 | CAVTTDSWGKLQF | TRBV6-5*01 | TRBJ2-7*01 | CASRPGLAGGRPEQYF | LLFGFPVYV |
| 4ftv | A*02 | TRAV12-2*01 | TRAJ24*02 | CAVTTDSWGKLQF | TRBV6-5*01 | TRBJ2-7*01 | CASRPGLMSAQPEQYF | LLFGYPVYV |
| 3hg1 | A*02 | TRAV12-2*01 | TRAJ27*01 | CAVNVAGKSTF | TRBV30*01 | TRBJ2-2*01 | CAWSETGLGTGELFF | ELAGIGILTV |
| 4eup | A*02 | TRAV12-2*01 | TRAJ45*01 | CAVSGGGADGLTF | TRBV28*01 | TRBJ2-1*01 | CASSFLGTGVEQYF | ALGIGILTV |
| 5c0c | A*02 | TRAV12-3*01 | TRAJ12*01 | CAMRGDSSYKLIF | TRBV12-4*01 | TRBJ2-4*01 | CASSLWEKLAKNIQYF | RQFGPDWIVA |
| 5eu6 | A*02 | TRAV21*01 | TRAJ53*01 | CAVLSSGGSNYKLTF | TRBV7-3*01 | TRBJ2-3*01 | CASSFIGGTDTQYF | YLEPGPVTV |
| 2p5e | A*02 | TRAV21*01 | TRAJ6*01 | CAVRPLLDGTYIPTF | TRBV6-5*01 | TRBJ2-2*01 | CASSYLGNTGELFF | SLLMWITQC |
| 2bnq | A*02 | TRAV21*01 | TRAJ6*01 | CAVRPTSGGSYIPTF | TRBV6-5*01 | TRBJ2-2*01 | CASSYVGNTGELFF | SLLMWITQV |
| 4mnq | A*02 | TRAV22*01 | TRAJ40*01 | CAVDSATALPYGYIF | TRBV6-5*01 | TRBJ1-1*01 | CASSYQGTEAFF | ILAKFLHWL |
| 5men | A*02 | TRAV22*01 | TRAJ40*01 | CAVDSATSGTYKYIF | TRBV6-5*01 | TRBJ1-1*01 | CASSYQGTEAFF | ILAKFLHWL |
| 5isz | A*02 | TRAV24*01 | TRAJ27*01 | CAFDTNAGKSTF | TRBV19*01 | TRBJ2-7*01 | CASSIFGQREQYF | GILGFVFTL |
| 5d2l | A*02 | TRAV24*01 | TRAJ49*01 | CAFITGNQFYF | TRBV7-2*02 | TRBJ2-5*01 | CASSQTQLWETQYF | NLVPMVATV |
| 3gsn | A*02 | TRAV24*01 | TRAJ49*01 | CARNTGNQFYF | TRBV6-5*01 | TRBJ1-2*01 | CASSPVTGGIYGYTF | NLVPMVATV |
| 5d2n | A*02 | TRAV26-2*01 | TRAJ43*01 | CILDNNNDMRF | TRBV7-6*01 | TRBJ1-4*01 | CASSLAPGTTNEKLFF | NLVPMVATV |
| 5euo | A*02 | TRAV27*01 | TRAJ37*02 | CAGAIGPSNTGKLIF | TRBV19*01 | TRBJ2-7*01 | CASSIRSSYEQYF | GILGFVFTL |
| 5hho | A*02 | TRAV27*01 | TRAJ42*01 | CAGAGSQGNLIF | TRBV19*01 | TRBJ2-7*01 | CASSIRSSYEQYF | GILEFVFTL |
| 2vlr | A*02 | TRAV27*01 | TRAJ42*01 | CAGAGSQGNLIF | TRBV19*01 | TRBJ2-7*01 | CASSSRASYEQYF | GILGFVFTL |
| 1oga | A*02 | TRAV27*01 | TRAJ42*01 | CAGAGSQGNLIF | TRBV19*01 | TRBJ2-7*01 | CASSSRSSYEQYF | GILGFVFTL |
| 1bd2 | A*02 | TRAV29/DV5*01 | TRAJ54*01 | CAAMEGAQKLVF | TRBV6-5*01 | TRBJ2-7*01 | CASSYPGGGFYEQYF | LLFGYPVYV |
| 5e6i | A*02 | TRAV35*01 | TRAJ37*02 | CAGPGGSSNTGKLIF | TRBV27*01 | TRBJ2-2*01 | CASSLIYPGELFF | GILGFVFTL |
| 3qeq | A*02 | TRAV35*01 | TRAJ49*01 | CAGGTGNQFYF | TRBV10-3*01 | TRBJ1-5*01 | CAISEVGVGQPQHF | AAGIGILTV |
| 4zez | A*02 | TRAV38-2/DV8*01 | TRAJ30*01 | CAYGEDDKIIF | TRBV25-1*01 | TRBJ2-7*01 | CASRRGPYEQYF | KLVALVINAV |
| 5jhd | A*02 | TRAV38-2/DV8*01 | TRAJ52*01 | CAWGVNAGGTSYGKLTF | TRBV19*01 | TRBJ1-2*01 | CASSIGVYGYTF | GILGFVFTL |

*Table 4 continued on next page*

*Table 4 continued*

| PDB ID* | HLA allele | V$\alpha$ | J$\alpha$ | CDR3$\alpha$ | V$\beta$ | J$\beta$ | CDR3$\beta$ | Peptide |
|---|---|---|---|---|---|---|---|---|
| 3o4l | A*02 | TRAV5*01 | TRAJ31*01 | CAEDNNARLMF | TRBV20-1*01 | TRBJ1-2*01 | CSARDGTGNGYTF | GLCTLVAML |
| 3vxs | A*24 | TRAV21*01 | TRAJ12*01 | CAVRMDSSYKLIF | TRBV7-9*01 | TRBJ2-2*01 | CASSSWDTGELFF | RYPLTLGWCF |
| 3vxm | A*24 | TRAV8-3*01 | TRAJ28*01 | CAVGAPSGAGSYQLTF | TRBV4-1*01 | TRBJ2-7*01 | CASSPTSGIYEQYF | RFPLTFGWCF |
| 3sjv | B*08 | TRAV12-1*01 | TRAJ23*01 | CVVRAGKLIF | TRBV6-2*01 | TRBJ2-4*01 | CASGQGNFDIQYF | FLRGRAYGL |
| 3ffc | B*08 | TRAV14/DV4*01 | TRAJ49*01 | CAMREDTGNQFYF | TRBV11-2*01 | TRBJ2-3*01 | CASSFTWTSGGATDTQYF | FLRGRAYGL |
| 1mi5 | B*08 | TRAV26-2*01 | TRAJ52*01 | CILPLAGGTSYGKLTF | TRBV7-8*01 | TRBJ2-7*01 | CASSLGQAYEQYF | FLRGRAYGL |
| 4qrp | B*08 | TRAV9-2*01 | TRAJ43*01 | CALSDPVNDMRF | TRBV11-2*01 | TRBJ1-5*01 | CASSLRGRGDQPQHF | HSKKKCDEL |
| 4g9f | B*27 | TRAV14/DV4*02 | TRAJ21*01 | CAMRDLRDNFNKFYF | TRBV6-5*01 | TRBJ1-1*01 | CASREGLGGTEAFF | KRWIIMGLNK |
| 4jrx | B*35 | TRAV19*01 | TRAJ34*01 | CALSGFYNTDKLIF | TRBV6-1*01 | TRBJ1-1*01 | CASPGETEAFF | LPEPLPQGQLTAY |
| 2ak4 | B*35 | TRAV19*01 | TRAJ34*01 | CALSGFYNTDKLIF | TRBV6-1*01 | TRBJ2-7*01 | CASPGLAGEYEQYF | LPEPLPQGQLTAY |
| 3mv7 | B*35 | TRAV20*01 | TRAJ58*01 | CAVQDLGTSGSRLTF | TRBV9*01 | TRBJ2-2*01 | CASSARSGELFF | HPVGEADYFEY |
| 4jry | B*35 | TRAV39*01 | TRAJ33*01 | CAVGGGSNYQLIW | TRBV5-6*01 | TRBJ2-7*01 | CASSRTGSTYEQYF | LPEPLPQGQLTAY |
| 3dxa | B*44 | TRAV26-1*01 | TRAJ13*02 | CIVWGGYQKVTF | TRBV7-9*01 | TRBJ2-1*01 | CASRYRDDSYNEQFF | EENLLDFVRF |
| 3kpr | B*44 | TRAV26-2*01 | TRAJ52*01 | CILPLAGGTSYGKLTF | TRBV7-8*01 | TRBJ2-7*01 | CASSLGQAYEQYF | EEYLKAWTF |
| 4mji | B*51 | TRAV17*01 | TRAJ22*01 | CATDDDSARQLTF | TRBV7-3*01 | TRBJ2-2*01 | CASSLTGGGELFF | TAFTIPSI |
| 2ypl | B*57 | TRAV5*01 | TRAJ13*01 | CAVSGGYQKVTF | TRBV19*01 | TRBJ1-2*01 | CASTGSYGYTF | KAFSPEVIPMF |
| 4p4k | DPA1*01/DPB1*352 | TRAV9-2*01 | TRAJ28*01 | CALSLYSGAGSYQLTF | TRBV5-1*01 | TRBJ2-5*01 | CASSLAQGGETQYF | QAFWIDLFETIG |
| 4may | DQA1*01/DQB1*05 | TRAV13-1*01 | TRAJ48*01 | CAASSFGNEKLTF | TRBV7-3*01 | TRBJ2-3*01 | CATSALGDTQYF | QLVHFVRDFAQL |
| 5ks9 | DQA1*03/DQB1*03 | TRAV20*01 | TRAJ39*01 | CAVALNNNAGNMLTF | TRBV9*01 | TRBJ2-3*01 | CASSVAPGSDTQYF | APSGEGSFQPSQENPQ |
| 4gg6 | DQA1*03/DQB1*03 | TRAV26-2*01 | TRAJ45*01 | CILRDGRGGADGLTF | TRBV9*01 | TRBJ2-7*01 | CASSVAVSAGTYEQYF | QQYPSGEGSFQPSQENPQ |
| 4z7u | DQA1*03/DQB1*03 | TRAV26-2*01 | TRAJ49*01 | CILRDRSNQFYF | TRBV9*01 | TRBJ2-5*01 | CASSTTPGTGTETQYF | APSGEGSFQPSQENPQGS |
| 4z7v | DQA1*03/DQB1*03 | TRAV26-2*01 | TRAJ54*01 | CILRDSRAQKLVF | TRBV9*01 | TRBJ2-7*01 | CASSAGTSGEYEQYF | APSGEGSFQPSQENPQGS |
| 4z7w | DQA1*03/DQB1*03 | TRAV8-3*01 | TRAJ36*01 | CAVGETGANNLFF | TRBV6-1*01 | TRBJ2-1*01 | CASSEARRYNEQFF | APSGEGSFQPSQENPQGS |
| 4ozh | DQA1*05/DQB1*02 | TRAV26-1*01 | TRAJ32*01 | CIVWGGATNKLIF | TRBV7-2*01 | TRBJ2-3*01 | CASSVRSTDTQYF | APQPELPYPQPGS |
| 4ozg | DQA1*05/DQB1*02 | TRAV26-1*01 | TRAJ45*01 | CIVLGGADGLTF | TRBV7-2*01 | TRBJ2-3*01 | CASSFRFTDTQYF | APQPELPYPQPGS |
| 4ozf | DQA1*05/DQB1*02 | TRAV26-1*01 | TRAJ54*01 | CIAFQGAQKLVF | TRBV7-2*01 | TRBJ2-3*01 | CASSFRALAADTQYF | APQPELPYPQPGS |

Table 4 continued

| PDB ID* | HLA allele | Vα | Jα | CDR3α | Vβ | Jβ | CDR3β | Peptide |
|---------|-----------|-----|-----|-------|-----|-----|-------|---------|
| 4ozi | DQA1*05/ DQB1*02 | TRAV4*01 | TRAJ4*01 | CLVGDGGSFSGGYNKLIF | TRBV20-1*01 | TRBJ2-5*01 | CSAGVGGQETQYF | QPFPQPELPYPGS |
| 5ksa | DQA1*05/ DQB1*03 | TRAV20*01 | TRAJ33*01 | CAVQFMDSNYQLIW | TRBV9*01 | TRBJ2-7*01 | CASSVAGTPSYEQYF | QPQQSFPEQEA |
| 5ksb | DQA1*05/ DQB1*03 | TRAV20*01 | TRAJ6*01 | CAVQASGGSYIPTF | TRBV9*01 | TRBJ2-3*01 | CASSNRGLGTDTQYF | GPQQSFPEQEA |
| 4e41 | DRA*01/ DRB1*01 | TRAV22*01 | TRAJ18*01 | CAVDRGSTLGRLYF | TRBV5-8*01 | TRBJ2-5*01 | CASSQIRETQYF | GELIGILNAAKVPAD |
| 2iam | DRA*01/ DRB1*01 | TRAV22*01 | TRAJ54*01 | CAALIQGAQKLVF | TRBV6-6*01 | TRBJ1-3*01 | CASTYHGTGYF | GELIGILNAAKVPAD |
| 1fyt | DRA*01/ DRB1*01 | TRAV8-4*01 | TRAJ48*01 | CAVSESPFGNEKLTF | TRBV28*01 | TRBJ1-2*01 | CASSSTGLPYGYTF | PKYVKQNTLKLAT |
| 3o6f | DRA*01/ DRB1*04 | TRAV26-2*01 | TRAJ32*01 | CTVYGGATNKLIF | TRBV20-1*01 | TRBJ1-6*01 | CSARGGSYNSPLHF | FSWGAEGQRPGFGSGG |
| 1j8h | DRA*01/ DRB1*04 | TRAV8-4*01 | TRAJ48*01 | CAVSESPFGNEKLTF | TRBV28*01 | TRBJ1-2*01 | CASSSTGLPYGYTF | PKYVKQNTLKLAT |
| 2wbj | DRA*01/ DRB1*15 | TRAV17*01 | TRAJ40*01 | CATDTTSGTYKYIF | TRBV20-1*01 | TRBJ2-1*01 | CSARDLTSGANNEQFF | MDFARVHFISALHGSGG |
| 4h1l | DRA*01/ DRB3*03 | TRAV8-3*01 | TRAJ37*01 | CAVGASGNTGKLIF | TRBV19*01 | TRBJ2-2*01 | CASSLRDGYTGELFF | QHIRCNIPKRISA |
| 1zgl | DRA*01/ DRB5*01 | TRAV9-2*01 | TRAJ12*01 | CALSGGDSSYKLIF | TRBV5-1*01 | TRBJ1-1*01 | CASSLADRVNTEAFF | VHFFKNIVTPRTPGG |

*If there are multiple structures with the same TCR and HLA allele, only the ID of the highest-resolution structure is given. During CDR3β contact analysis, however, we combined the contacts from all redundant structures, downweighting so as to equalize the contribution from all TCR/HLA pairs.

DOI: https://doi.org/10.7554/eLife.38358.033

information across loci using trans-locus sequence alignments. For class I we used the merged alignment (ClassI_prot.txt) available from the IPD-IMGT/HLA (*Robinson et al., 2015*) database. Starting with multiple sequence alignments for individual class II loci from the IPD-IMGT/HLA database, we inserted gaps as needed in order to created merged alignments for the class II $\alpha$ and $\beta$ chains. These alignments provided a common reference frame in which to combine residue-residue contacts from the TCR:peptide-MHC structures. We considered two amino acid residues to be in contact if they had a side chain heavyatom contact distance less than or equal to 4.5Å. The CDR3β contact frequency for an alignment position (class I, class II-$\alpha$, or class II-$\beta$) was defined to be the total number of contacted CDR3β amino acids observed for that position, divided by the total number of structures analyzed. Redundancy in the structural database was assessed at the level of TCR and HLA sequence, ignoring the sequence of the peptide. Contacts from a set of $n$ structures all containing the same TCR and HLA were given a weight of $1/n$ when computing the residue contact frequencies. The statistical significance of correlations between HLA allele charge and average HLA-associated TCR CDR3 charge were computed using a 2-sided test as implemented in the function scipy.stats.linregress.

## Software availability

C++ source code implementing the clustering, generation probability, and correlation algorithms described here is available at https://github.com/phbradley/pubtcrs (copy archived at https://github.com/elifesciences-publications/pubtcrs [*Bradley, 2018*]).

## Acknowledgements

This work was supported in part through the NIH/NCI Cancer Center Support Grant P30 CA015704 and by NIH NHLBI grant R01-HL105914 to JH, as well as R01 GM113246 and U19 AI117891. The research of Frederick Matsen was supported in part by a Faculty Scholar grant from the Howard

Hughes Medical Institute and the Simons Foundation. We gratefully acknowledge superlative computing support from Fred Hutch scientific computing and thank Paul Thomas and Jeremy Crawford for helpful comments on a preliminary version of this manuscript.

## Additional information

### Funding

| Funder | Grant reference number | Author |
|---|---|---|
| National Institutes of Health | CA015704 | Anajane Smith<br>Gary Schoch<br>John A Hansen<br>Frederick A Matsen IV<br>Philip Bradley |
| National Institutes of Health | R01-HL105914 | Anajane Smith<br>Gary Schoch<br>John A Hansen |
| National Institutes of Health | R01-GM113246 | Frederick A Matsen IV |
| National Institutes of Health | U19-AI117891 | Frederick A Matsen IV |
| Howard Hughes Medical Institute | 55108544 | Frederick A Matsen IV |
| Fred Hutchinson Cancer Research Center | Salary support | Philip Bradley |

The funders had no role in study design, data collection and interpretation, or the decision to submit the work for publication.

### Author contributions

William S DeWitt III, Conceptualization, Data curation, Methodology, Writing—original draft, Writing—review and editing; Anajane Smith, Gary Schoch, Data curation, Writing—review and editing; John A Hansen, Data curation, Supervision, Funding acquisition, Writing—review and editing; Frederick A Matsen IV, Conceptualization, Supervision, Funding acquisition, Methodology, Writing—original draft, Writing—review and editing; Philip Bradley, Conceptualization, Resources, Data curation, Software, Formal analysis, Supervision, Funding acquisition, Validation, Investigation, Visualization, Methodology, Writing—original draft, Project administration, Writing—review and editing

### Author ORCIDs

William S DeWitt III  http://orcid.org/0000-0002-6802-9139
Frederick A Matsen IV  http://orcid.org/0000-0003-0607-6025
Philip Bradley  http://orcid.org/0000-0002-0224-6464

### Ethics

Human subjects: All samples were collected and analyzed, and informed consent and consent to publish were obtained, according to research protocols approved by the Fred Hutchinson Cancer Research Center (FHCRC) Institutional Review Board.

### Decision letter and Author response

Decision letter https://doi.org/10.7554/eLife.38358.042
Author response https://doi.org/10.7554/eLife.38358.043

## Additional files

### Supplementary files

• Transparent reporting form
DOI: https://doi.org/10.7554/eLife.38358.034

## Data availability

Data and analysis scripts needed to reproduce the findings of this study have been deposited in the Zenodo database (doi:10.5281/zenodo.1248193).

The following dataset was generated:

| Author(s) | Year | Dataset title | Dataset URL | Database, license, and accessibility information |
|---|---|---|---|---|
| Philip Bradley | 2018 | Supporting dataset for the publication 'Human T cell receptor occurrence patterns encode immune history, genetic background, and receptor specificity' | http://dx.doi.org/10.5281/zenodo.1248193 | Publicly available at Zenodo (https://zenodo.org/) |

The following previously published datasets were used:

| Author(s) | Year | Dataset title | Dataset URL | Database, license, and accessibility information |
|---|---|---|---|---|
| Emerson RO, De-Witt WS, Vignali M, Gravley J, Hu JK, Osborne EJ, Desmarais C, Klinger M, Carlson CS, Hansen JA, Rieder M, Robins HS | 2017 | Immunosequencing identifies signatures of cytomegalovirus exposure history and HLA-mediated effects on the T cell repertoire | https://doi.org/10.21417/B7001Z | Publicly available in Adaptive Biotechnology's ImmuneAccess database |
| Britanova OV, Shugay M, Merzlyak EM, Staroverov DB, Putintseva EV, Turchaninova MA, Mamedov IZ, Pogorelyy MV, Bolotin DA, Izraelson M, Davydov AN, Egorov ES, Kasatskaya SA, Rebrikov DV, Lukyanov S, Chudakov DM | 2016 | Dynamics of Individual T Cell Repertoires: From Cord Blood to Centenarians | https://www.ncbi.nlm.nih.gov/bioproject/PRJNA316572/ | Publicly available at NCBI Short Read Archive (accession no. PRJNA316572) |

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
