## [Decision Letter]

Thank you for submitting your article "Human T cell receptor occurrence patterns encode immune history, genetic background, and receptor specificity" for consideration by *eLife*. Your article has been reviewed by Arup Chakraborty as the Senior Editor, a Reviewing Editor, and three reviewers. The following individuals involved in review of your submission have agreed to reveal their identity: Yuval Elhanati (Reviewer #1); Bram Gerritsen (Reviewer #2).

The reviewers have discussed the reviews with one another and the Reviewing Editor has drafted this decision to help you prepare a revised submission.

Summary:

The paper develops a statistical framework to study the links between TCR co-occurrence patterns, HLA-association strength, and TCR sequence similarity based on the previously published dataset of Emerson et al., 2017. This is an advanced analysis that goes beyond simple V-D-J rearrangement statistics, corrects for baselines and identifies correlations between HLA type and TCR motifs.

The reviewers and the editor agree that the paper develops a statistically sound framework, which leads to interesting insights. However, the reviewers raised a number of comments that must be addressed before publication. Specifically, the experts in the field found the paper difficult to read and often unclear. The combination of previously known results and facts and new results make it hard to capture the main finding and the main new results of the study. The authors should also discuss potential biases that may influence their result.

Essential revisions:

1) A significant concern with the paper is related to readability. Especially for the non-expert, the paper is heavy with specific terms and acronyms and are difficult to follow. And even to researchers in the field, the long paragraphs detailing the analysis are hard to parse.

I suggest to try and break down the analysis more, and using some more clear definitions in the main text. Especially useful might be a cartoon or chart, displaying the connection between the occurrence matrix M, the HLA alleles and the clustering analysis. It will be very helpful to "see" the main connections used in the paper such as P_{CO}, P_{HLA} and the clusters.

It's also hard to follow the different stages of the analysis with similar names, such as HLA-associated TCRs and HLA restricted clustering. Maybe this can be better explained in the outline.

2) The bulk of the analysis focused on a small fraction (about 1:1000) strongly HLA-associated TCRs found. That only few TCRs ended up in clusters might be due to the strict selection criteria (FWER 0.05, DBSCAN parameters) and/or limitations of the dataset, such as unsorted cells, lacking an α chain, and insufficient sequencing depth. Might many TCRs be missed because different HLA-alleles may display the same antigen (weakening the HLA-association)? Given how restrictive HLA-association seems to be, causing many expanded TCRs to be missed, it might be interesting to cluster on expansion index (I_exp) and co-occurrence. This way more of the TCRs in the dataset could be informative for immune history.

3) Several critical factors, such as the extent of the influence of common pathogen exposure on the observed HLA-mediated alterations of the TCR repertoire and a complete lack of TCRα sequencing data, can bias the results and require re-considering the set of conclusions reported. The presence of both HLA and common pathogen (CMV in the latter case) imprinting was previously reported with exactly the same dataset in Emerson et al., 2017. The authors focus their study on HLA-restricted TCRs, obtaining results that follow in an obvious way from the Emerson et al. study. For example, it has been clear for some time by sorting and sequencing antigen-specific populations that MHC restriction is detrimental in obtaining specific TCRs from a given donor. Moreover, the association between MHC alleles and TCR sequences was previously described in Sharon et al., 2016. The authors should update their manuscript to make clear the novel findings of their study and separate them from miscellaneous observations and replications of previous results.

4) The authors use unsorted T-cells, so one would expect that the most evident process that shapes TCR repertoires is the HLA restriction of response to common pathogens. For examples, almost 95% of population is positive for the Epstein-Barr virus, meaning that a trace of EBV epitopes restricted to MHCI and MHCII alleles will dominate across the observed repertoire. If a donor doesn't have an HLA allele to present a given epitope there will be no corresponding clonal expansion. Less studied (and thus more interesting) aspects of the TCR repertoire formation, such as thymic selection in the context of specific HLAs and the influence of individual V/J allelic variants, are not covered.

5) Grouping TCRs by associated pathogens is misleading in the context of the analysis of HLA-restricted repertoires. In case two epitopes are presented by the same HLA and come from different species it is more reasonable to group TCRs by them, rather than group TCRs specific to a pair of epitopes presented by distinct HLAs and coming from the same species. Authors should be more precise when annotating the set of TCRs in case they analyze the effects of MHC restriction.

6) The authors do not consider TCRα chain sequences for most of their analysis. However, they may be detrimental for explaining MHC restriction effects in a large number of settings, for example:

- Recent studies such as Culshaw et al., 2017 and Cole et al., 2009 show a substantial germline bias coming from Vα segment usage patterns in antigen-specific responses that allow a large set of TCRβ sequences.

- The inference of MAIT TCR clusters is trivial given TCRα sequencing data.

- Authors report on TCRs that show no MHC association and state that "TCRs with HLA promiscuity may be especially interesting from a diagnostic perspective, since their phenotype associations may be more robust to differences in genetic background.". These TCRs may be strongly associated with MHC via their α chain.

The authors should show, based on available datasets (e.g. PairSEQ by Howie et al., and single-cell data), that their findings hold when considering the pairing of TCRβ chain sequences with several potential TCRα chain sequences.

7) The authors also ignore the fact that they operate with populations that were not sorted for CD4/8 markers. In fact, in current setting authors cannot show that a given TCR that is found to be associated with a given MHCI/II allele originates from either T-killer or T-helper cells. Authors should estimate (e.g. by comparing their associated sequences with known CD8/CD4 variants) how many of their MHC-associated TCRs are misattributed to wrong MHC class. The authors only find a significant association with MHC class II β chain position 70. Given the fact that they've checked various MHC class I and MHC class II α positions, will the significance hold when applying multiple testing correction? Otherwise, the result may be less general than the results reported by Sharon et al., 2016 and expected to be encountered by chance given the number of tested MHC residues.

---

## [Author Response]

Summary:The paper develops a statistical framework to study the links between TCR co-occurrence patterns, HLA-association strength, and TCR sequence similarity based on the previously published dataset of Emerson et al., 2017. This is an advanced analysis that goes beyond simple V-D-J rearrangement statistics, corrects for baselines and identifies correlations between HLA type and TCR motifs.The reviewers and the editor agree that the paper develops a statistically sound framework, which leads to interesting insights. However, the reviewers raised a number of comments that must be addressed before publication. Specifically, the experts in the field found the paper difficult to read and often unclear. The combination of previously known results and facts and new results make it hard to capture the main finding and the main new results of the study. The authors should also discuss potential biases that may influence their result.We have endeavored to make the paper clearer, guided by the reviewers' suggestions. We have added a new introductory figure (Figure 1) that provides an overview of the main results in a graphical format and a new methods figure (Figure 12) that illustrates the co-occurrence analysis. We have also added additional explanatory text throughout the manuscript. Finally, we have added additional discussion of potential biases to the Discussion section.Essential revisions:1) A significant concern with the paper is related to readability. Especially for the non-expert, the paper is heavy with specific terms and acronyms and are difficult to follow. And even to researchers in the field, the long paragraphs detailing the analysis are hard to parse.I suggest to try and break down the analysis more, and using some more clear definitions in the main text. Especially useful might be a cartoon or chart, displaying the connection between the occurrence matrix M, the HLA alleles and the clustering analysis. It will be very helpful to "see" the main connections used in the paper such as P_{CO}, P_{HLA} and the clusters.

This is a great suggestion which we have tried to follow in our two new figures.

It's also hard to follow the different stages of the analysis with similar names, such as HLA-associated TCRs and HLA restricted clustering. Maybe this can be better explained in the outline.

We hope that our overview figures and in-text explanations have clarified these points.

2) The bulk of the analysis focused on a small fraction (about 1:1000) strongly HLA-associated TCRs found. That only few TCRs ended up in clusters might be due to the strict selection criteria (FWER 0.05, DBSCAN parameters) and/or limitations of the dataset, such as unsorted cells, lacking an α chain, and insufficient sequencing depth. Might many TCRs be missed because different HLA-alleles may display the same antigen (weakening the HLA-association)?

This is an excellent question that we have continued to explore throughout the project: to what extent are (some) TCRs capable of seeing epitopes in the context of multiple HLA alleles, and can we see signal for such promiscuity in bulk datasets? We have certainly been able to find individual TCRs which are strongly associated with two different HLA alleles. Generally, these are alleles that differ only at 4-digit resolution (e.g. DRB1*04:01 and DRB1*04:04), although there are rare cases where the alleles are more diverged. The total number of such cases is relatively small, which makes us think that this may not be a major cause for missing functional TCRs. For example, when we compare, for each TCR, the p-value of its most strongly associated 2-digit allele (e.g., A*02) with the p-value of its most strongly-associated 4-digit allele (e.g., A*02:01), the 4-digit allele associations are generally stronger. We have also looked at TCR associations with 'clades' of related HLA alleles defined by sequence similarity, and the total number and set of TCRs associated with these clades was largely concordant with the canonical HLA-allele associated TCRs we report in the manuscript. Of course, if a TCR can see epitope presented by many alleles then its association with any single allele will be fairly weak, which would make it harder to detect. On the other hand, our analysis of HLA-association for CMV-associated TCRs (Figure 9) suggests that most of those CMV-responsive TCRs are restricted by a single HLA allele, which argues against large-scale HLA promiscuity, at least for pathogen associated TCR chains. We have modified the discussion to address this question.

Given how restrictive HLA-association seems to be, causing many expanded TCRs to be missed, it might be interesting to cluster on expansion index (I_exp) and co-occurrence. This way more of the TCRs in the dataset could be informative for immune history.

This is an excellent suggestion and indeed we explored a number of variants of this idea (filtering or stratifying TCR chains by degree of expansion) in our initial analyses. We generally got results similar to those from global clustering (Figure 3), in which HLA-associated clusters dominated with only a few additional non-HLA associated clusters. That said, there is definitely more to be done along these lines and we have added a discussion of extensions and alternative approaches to the discussion. In the interests of maintaining a manageable focus for this manuscript, however, we have elected not to add a new clustering analysis at this time.

3) Several critical factors, such as the extent of the influence of common pathogen exposure on the observed HLA-mediated alterations of the TCR repertoire and a complete lack of TCRα sequencing data, can bias the results and require re-considering the set of conclusions reported. The presence of both HLA and common pathogen (CMV in the latter case) imprinting was previously reported with exactly the same dataset in Emerson et al., 2017. The authors focus their study on HLA-restricted TCRs, obtaining results that follow in an obvious way from the Emerson et al. study.

Here are some of the key differences between our study and the analysis of Emerson et al.

-The new HLA data that we report allow us to find many new significantly HLA-associated TCRs (e.g., we find more than 50,000 using the same P-value threshold at which Emerson et al. identify ~15,000) and to make systematic comparisons across loci.

-Emerson et al., do not analyze the sequences of their HLA associated TCRs at all. So, for example our result that 8 of the top 10 A*02:01-associated TCRs appear specific for just two pathogen epitopes (Influenza M158 and EBV BMLF1) goes beyond the results they report.

-Emerson et al. do not analyze TCR-TCR co-occurrence at all. Thus, our results dealing with clustering of TCRs by co-occurrence seem novel.

-Emerson et al. do not analyze HLA-CDR3 covariation nor do they investigate the degree of HLA restriction of CMV-associated TCRs, as we do.

-Our results on the distribution of generation probabilities and clonal expansion indices of HLA-associated TCRs have no parallel in the Emerson et al. study.

For example, it has been clear for some time by sorting and sequencing antigen-specific populations that MHC restriction is detrimental in obtaining specific TCRs from a given donor. Moreover, the association between MHC alleles and TCR sequences was previously described in Sharon et al., 2016.

The novel aspect here is that we are finding covariation with the CDR3 sequence rather than the V gene. This is somewhat surprising given the textbook picture of the V-region CDR1 and CDR2 loops contacting MHC and the CDR3 interacting primarily with peptide. We have attempted to clarify this in the section on HLA-CDR3 covariation.

The authors should update their manuscript to make clear the novel findings of their study and separate them from miscellaneous observations and replications of previous results.

We have attempted to do so.

4) The authors use unsorted T-cells, so one would expect that the most evident process that shapes TCR repertoires is the HLA restriction of response to common pathogens. For examples, almost 95% of population is positive for the Epstein-Barr virus, meaning that a trace of EBV epitopes restricted to MHCI and MHCII alleles will dominate across the observed repertoire. If a donor doesn't have an HLA allele to present a given epitope there will be no corresponding clonal expansion.

We agree that the general observation that pathogen responses shape the TCR repertoire is not surprising or novel. We would argue that the degree and specific nature of the imprint of pathogen responses that we discover is novel and in many cases non-obvious. For example, we would not have predicted beforehand that in A*24:02 positive individuals, the dominant cluster of TCRs would be responding to parvovirus B19 rather than, say, EBV or Influenza. Or that the top 2 HLA-associated TCRs overall would be canonical influenza and EBV-associated sequences, rather than TCRs responding for example to common self-antigens (cf. the recent study of Madi et al., 2017). Or that 8 of the top 10 HLA-associated TCRs would be attributable to just two epitopes.

Less studied (and thus more interesting) aspects of the TCR repertoire formation, such as thymic selection in the context of specific HLAs and the influence of individual V/J allelic variants, are not covered.

In fact, we did discover a TCR cluster driven by J-allelic variation (Figure

3). We agree that teasing apart the influence of HLA on thymic selection is an important and interesting goal. This large dataset may not be the best place to do that, given that the cells are unsorted, and the sequencing may not be as deep as one would like to investigate trends in the naive pool.

5) Grouping TCRs by associated pathogens is misleading in the context of the analysis of HLA-restricted repertoires. In case two epitopes are presented by the same HLA and come from different species it is more reasonable to group TCRs by them, rather than group TCRs specific to a pair of epitopes presented by distinct HLAs and coming from the same species. Authors should be more precise when annotating the set of TCRs in case they analyze the effects of MHC restriction.

The reviewer's suggestion is not entirely clear to us. In our analysis of HLA-restricted repertoires we do in fact group TCRs first by their restricting HLA when looking for clusters of co-occurring TCRs. The assignment to inferred pathogens is done only at the end as an annotation step and does not affect the cluster composition. When seeking an explanation for a group of co-occurring receptor chains, it seems natural to look for a response to a shared pathogen, since this immune exposure could explain the tendency of the receptors to co-occur across the cohort.

6) The authors do not consider TCRα chain sequences for most of their analysis.

The lack of α chain sequences is an unfortunate limitation of the dataset, one that we now emphasize more strongly in the Discussion section of the manuscript. We are not aware of α chain repertoire data for any similarly-sized cohorts with parallel HLA typing.

However, they may be detrimental for explaining MHC restriction effects in a large number of settings, for example:- Recent studies such as Culshaw et al., 2017 and Cole et al., 2009 show a substantial germline bias coming from Vα segment usage patterns in antigen-specific responses that allow a large set of TCRβ sequences.

This is a good point which we now touch on in the Discussion section. Indeed, we observed strong α chain bias for several of the repertoires in our recent study of epitope-specific repertoires (Dash et al., 2017). In fact, the EBV BMLF1 epitope is one for which the α preference (e.g., TRAV5) is stronger than the β chain preference. Yet it is notable that we are able to see a strong BMLF1 response in the A*02:01-associated TCRβ chain sequences.

- The inference of MAIT TCR clusters is trivial given TCRα sequencing data.

Agreed, the identification of MAIT TCRs *by sequence* is much more straightforward from α chain data. What is intriguing about our MAIT cluster is that it was defined based on *co-occurrence* across the cohort without using the receptor sequences at all. This suggests that there are subject-specific features (for example, subject age) that determine the frequency of MAIT cells in the blood, and that the influence of these features is strong enough to enable us to identify MAIT cells from their β-chain co-occurrence patterns alone.

- Authors report on TCRs that show no MHC association and state that "TCRs with HLA promiscuity may be especially interesting from a diagnostic perspective, since their phenotype associations may be more robust to differences in genetic background.". These TCRs may be strongly associated with MHC via their α chain.

This is an interesting point. We do indeed report a few TCRβ chains that show little or no MHC association and yet are strongly associated with CMV. It is possible that if we knew the paired α chains for these β chains, we would see that the specific paired alphas depend strongly on the HLA restriction. This wouldn't change the fact that the β chains themselves are not HLA associated, hence retain their CMV-predictive power independent of subject HLA type.

On the other hand, a predictor based on paired TCRs constructed from these chains might indeed be HLA-restricted in its applicability.

The authors should show, based on available datasets (e.g. PairSEQ by Howie et al., and single-cell data), that their findings hold when considering the pairing of TCRβ chain sequences with several potential TCRα chain sequences.

To the extent that our findings are about TCRβ chains and their cooccurrence and HLA association, they are independent of the specific α chain partners for these chains and will hold regardless. General conclusions about the strength of TCR-HLA association across loci and about the imprint of common pathogens could indeed be refined by paired data, and we have added discussion of this limitation. Unfortunately, we are not aware of large α-chain repertoires or pairSEQ datasets that would let us address this concern conclusively. The pairSEQ data of Howie et al. only covers a small number of samples, with deep coverage for just two individuals (subjectX and subjectY), and HLA typing is not publicly available for these individuals. For exploration of the MAIT cluster this was less of a concern as these cells are not HLA-restricted. Nonetheless, we followed the reviewer's suggestion and looked for paired chains for all of the TCRs in the HLA-restricted clusters, and we now provide this data as a supplementary source dataset for Figure 6. As expected, the number of clusters with multiple matched chains is rather limited, but for specific alleles (for example B*44-- it looks like both subjectX and subjectY are B*44 positive) we found some interesting trends. Specifically, we developed a probabilistic score to assess whether the α chains associated to a cluster are more similar to one another than would be expected by chance (provided as a column in the source datafile mentioned above), and several of the clusters show highly significant sequence clustering. This information could be helpful for further exploration of these clusters, as it provides candidate full-chain TCRs for synthesis and characterization. We appreciate the reviewer's advice in this regard and would welcome further suggestions for incorporating α chain analysis.

7) The authors also ignore the fact that they operate with populations that were not sorted for CD4/8 markers. In fact, in current setting authors cannot show that a given TCR that is found to be associated with a given MHCI/II allele originates from either T-killer or T-helper cells. Authors should estimate (e.g. by comparing their associated sequences with known CD8/CD4 variants) how many of their MHC-associated TCRs are misattributed to wrong MHC class.

This is an excellent suggestion. We have performed this analysis (Figure 5—figure supplement 1) and feel that it strengthens the manuscript. We also added a comparison between our TCR:HLA allele associations and the TCR:HLA allele pairings given in the VDJdb database and found that they are largely in agreement (both assessments can be found in the "HLA-associated TCRs" section).

The authors only find a significant association with MHC class II β chain position 70. Given the fact that they've checked various MHC class I and MHC class II α positions, will the significance hold when applying multiple testing correction? Otherwise, the result may be less general than the results reported by Sharon et al., 2016 and expected to be encountered by chance given the number of tested MHC residues.

This is a good suggestion. We looked only at the 7 positions with high contact frequencies listed in Table 2, so there are not a large number of comparisons, but we now report Bonferroni-corrected p-values in the text. Again, we are looking here at MHC-CDR3 covariation rather than MHC-V-gene covariation as done in Sharon et al.